# Is Exchangeability better than I.I.D. to handle Data Distribution Shifts while Pooling Data for Data-scarce Medical image segmentation?

**Ayush Roy**[1]                                                   AROY25@BUFFALO.EDU

**Samin Enam**[1]                                               SAMINENA@BUFFALO.EDU

**Jun Xia**[1]                                                        JUNXIA@BUFFALO.EDU

**Won Hwa Kim**[2]                                          WONHWA@POSTECH.AC.KR

**Vishnu Suresh Lokhande**[1]                          VISHNULO@BUFFALO.EDU

[1] *University at Buffalo (SUNY)*

[2] *Pohang University of Science and Technology (POSTECH)*

**Editors:** Accepted at MIDL 2026

## Abstract

Data scarcity is a major challenge in medical imaging, particularly for deep learning models. While data pooling (combining datasets from multiple sources) and data addition (adding more data from a new dataset) have been shown to enhance model performance, they are not without complications. Specifically, increasing the size of the training dataset through pooling or addition can induce distributional shifts, negatively affecting downstream model performance, a phenomenon known as the "Data Addition Dilemma". While the traditional i.i.d. assumption may not hold in multi-source contexts, assuming exchangeability across datasets provides a more practical framework for data pooling. In this work, we investigate medical image segmentation under these conditions, drawing insights from causal frameworks to propose a method for controlling foreground-background feature discrepancies across all layers of deep networks. This approach improves feature representations, which are crucial in data-addition scenarios. Our method achieves state-of-the-art segmentation performance on histopathology and ultrasound images across five datasets, including a novel ultrasound dataset that we have curated and contributed. Qualitative results demonstrate more refined and accurate segmentation maps compared to prominent baselines across three model architectures. The code is available on Github.

**Keywords:** List of keywords, comma separated.

## 1. Introduction

Medical imaging datasets often suffer from limited sample sizes due to budget constraints and strict study criteria, including genetic risk factors. This scarcity is further compounded by the lack of diagnostic labels, posing challenges for deep learning models that rely on supervised learning. Small datasets amplify the risk of models learning spurious correlations (Thompson et al., 2014; Lokhande et al., 2022), while distributional disparities hinder generalization to real-world clinical settings. In addition to that, smaller number of training samples lead to data memorization, data interpolation, and high variance models (Nakkiran et al., 2021; Power et al., 2022; Lin et al., 2023; Ying, 2019), all causing poor generalization. Though deep learning advancements show promise, issues of data quality and distribution mismatches remain significant barriers (Moyer et al., 2018). Semi-supervised learning

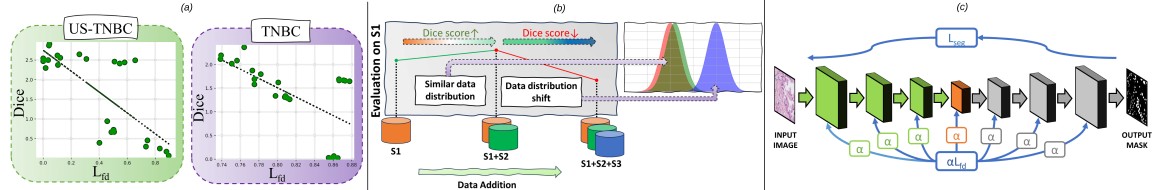

Figure 1: (a) **Strong correlation between Dice and $\mathcal{L}_{\mathbf{fd}}$ (foreground-background feature discrepancy loss).** Strong correlation in both NucleiSegNet decoder and CMUNet encoder layers for ultrasound and histopathology images. (b) **Impact of Data Distribution Shift on Model Performance.** Adding S2 (similar distribution) to S1 training improves S1 test Dice, as expected with more data. However, adding S3 (distribution shift) degrades performance, consistent with (Shen et al., 2024). (c) **Proposed $\mathcal{L}_{\mathbf{fd}}$ applied to all U-Net layers.** Encoder (green), Decoder (grey), and Bottleneck (orange) features represent mediator $Z$, optimized by $\mathcal{L}_{\mathbf{fd}}$. Each layer uses $\mathcal{L}_{\mathbf{fd}}$ with a unique learnable parameter $\alpha$.

and data augmentation provide partial solutions with varying effectiveness (Chapelle et al., 2006). Pooling data from multiple sites, combined with techniques like covariate matching and meta-analysis, has improved model robustness and generalization (Lokhande et al., 2022).

**Limitations of Data Augmentation in Medical Imaging.** Data augmentation techniques like rotations, flips, and crops generate synthetic training samples to improve model robustness (Carmon et al., 2019), but in medical imaging they often introduce clinically unrealistic artifacts. Flipping or cropping brain images can disrupt inherent anatomical asymmetries critical for diagnosis (Akash et al., 2021; Mehta et al., 2023), while such methods also fail to preserve realistic object boundaries (e.g., tumor margins) and spatial relationships, limiting their utility in semantic segmentation (Oliver et al., 2018; Goceri, 2023). **Alternatives: Data Pooling and Data Addition.** Given augmentation's limitations, two alternatives emerge: data pooling and data addition. Pooling aggregates multi-institutional datasets to enhance statistical power and diversity, but faces distributional shifts (scanner variations, population differences) and non-i.i.d. data structures (Fig. 1b), requiring harmonization algorithms (Moyer et al., 2020; Lokhande et al., 2022; Roy

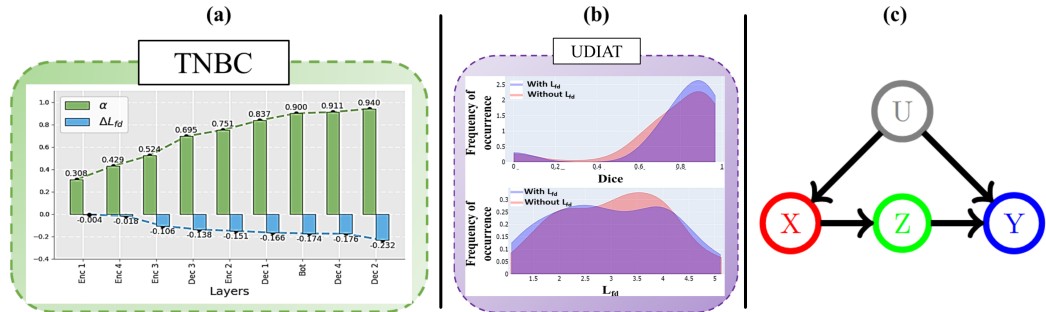

Figure 2: (a) $\alpha$ (layer-wise weights) vs $\mathcal{L}_{\mathbf{fd}}$ (feature discrepancy loss) for NucleiSegNet layers (TNBC) shows a similar trend across all models and datasets. (b) Right and left shifts in the test sample distribution for Dice scores and $\mathcal{L}_{\mathbf{fd}}$ after applying $\mathcal{L}_{\mathbf{fd}}$ (orange curve) for CMUNet (UDIAT), with a similar trend across datasets. Refined activation maps justify this improvement in Dice scores (see Figure 8, Dec 4 and Bot) after penalizing foreground-background discrepancy with $\mathcal{L}_{\mathbf{fd}}$. (c) Causal graph linking input X, mediator Z, label Y, and unobserved confounders U.

et al., 2026). Data addition incrementally integrates new data into pre-trained models, requiring adaptation strategies to reconcile distribution mismatches. However, the "data addition dilemma" (Shen et al., 2024) reveals that expanding training data can paradoxically degrade performance due to unresolved distribution shifts, underscoring the need for robust incremental learning strategies. **Causality-Driven Approaches.** To address distribution shifts and annotation biases in medical imaging (Sec. 1–2), causality-driven frameworks (Castro et al., 2020) are ideal for segmentation tasks in breast cancer and Alzheimer's Disease (AD). Breast cancer (most prevalent in women) causes over 14,000 annual deaths in Algeria alone (Lagree et al., 2021; aps, 2020), with detection relying on imaging to identify lesions (Evain et al., 2021; Touami and Benamrane, 2021), while AD diagnosis hinges on quantifying tau protein aggregates. Current quantification methods are labor-intensive and error-prone, necessitating automation.

Traditional segmentation models often fail to generalize due to unobserved confounders (scanner artifacts, anatomical variability) that corrupt the causal relationship between images X and annotations Y. Causal inference provides a principled framework (Schölkopf et al., 2012; Pearl, 2009; Bareinboim and Pearl, 2016). As in Fig. 2c, confounders $U$ (imaging protocols, demographics) influence both $X$ and $Y$, introducing spurious correlations. We adopt frontdoor adjustment (Pearl, 2009) using mediator $Z$ (foreground-background feature discrepancy) to disentangle causal effects. Our Feature Discrepancy Loss $L_{fd}$ operationalizes this by enhancing $Z$'s robustness to $U$, minimizing distributional shifts across datasets, ensuring $Y$ depends causally on $X$ rather than confounders. This addresses the "data addition dilemma" (Sec. 2) by stabilizing features during incremental learning. Selection bias from noise in $X$ that distorts $X \to Y$ (Castro et al., 2020) is mitigated, while label noise in $Y$ remains a separate challenge handled via label correction methods, enabling generalization in real-world clinical settings with unobserved biases. [1]

**Contributions.** This paper focuses on the segmentation task in medical imaging, a field that *still* presents significant challenges in accurately delineating complex anatomical structures and pathologies (Malhotra et al., 2022). Moreover, we leverage smaller models like UNet for our analysis. They remain sufficient in medical imaging due to their ability to accurately segment with limited data and minimal reliance on prompts or extensive fine-tuning (Yousef et al., 2023; Ahmadi et al., 2023). Our contributions stem from the observation that the Dice Score, a commonly used metric for evaluating segmentation quality, correlates with the discrepancy between foreground and background features in the intermediate representations generated by neural networks. This observation holds true across ultrasound and histopathology images (Fig 1 (a)), prompting the question: *Can controlling for foreground-background feature discrepancy improve the quality of these representations and, consequently, the Dice Score?* We show that it does. Specifically, we propose: **(a)** feature discrepancy loss to enhance feature distinction, reducing over- and under-segmentation in homogeneous pixel distributions; **(b)** theoretical bound showing that the negative logarithm of the Dice coefficient serves as a lower bound for the feature discrepancy loss, ensuring improved Dice scores when optimizing for this loss; **(c)** theoretical proof demonstrating the fact that the proposed feature discrepancy loss constrains the magnitude of the UNet layer weights, preventing the formation of a high-variance model prone to data memorization, a

---

1. A comprehensive review of related works is provided in Supplementary Sec. A

common issue in medical imaging datasets prone to limited samples; **(d)** introduction of a new ultrasound breast cancer dataset focused on triple-negative breast cancer (TNBC); and **(e)** a causal approach to address dataset distribution shift when integrating data from multiple sources. We achieve better segmentation performance across five datasets and significantly improve the segmentation performance of three prominent architectures.

## 2. Method

Causal diagrams formalize assumptions about data generation, improving model robustness and generalization to clinical data, which enhances diagnostic tools. Causal reasoning helps address data scarcity by analyzing cause-effect relationships. In our study of medical images $(X)$ and their corresponding segmentation ground truth $(Y)$, we explore the causal relationship between them. The relationship may be causal $(X \rightarrow Y)$, indicating $Y$ depends on $X$, or anticausal $(Y \rightarrow X)$, predicting the cause from the effect. The task is to estimate $P(Y \mid X)$. Manual segmentation is influenced by image content, resolution, contrast, and annotator understanding, thus suggesting a causal model, $X \rightarrow Y$.

**Axiom 1 (Modularity for X → Y):** *In the causal graph where X causes Y, intervening on X changes only the mechanism determining X, while the mechanism determining Y given X remains invariant.*

Axiom 1 indicates that $P(X)$ offers minimal information compared to $P(Y \mid X)$, implying that data augmentation and semi-supervised learning techniques are theoretically inadequate for resolving the data scarcity issue. A model trained on image-derived annotations will mainly reproduce the manual annotation process instead of predicting a pre-imaging ground truth, like the 'true' anatomy. While efforts to enhance data augmentation techniques for segmentation tasks continue (Yellapragada et al., 2024), our approach emphasizes utilizing existing data to improve segmentation outcomes.

### 2.1. Handling Data Scarcity through Causal Mediation

In data-scarce scenarios, the effective utilization of available samples is crucial. One approach focuses on improving the performance of underperforming samples, aligning with the Rawlsian principle (Lundgard, 2020) of prioritizing the worst-off samples. While techniques like upweighting show promise, they are impractical here due to the challenge of estimating reliable probability distributions in medical imaging datasets. To address this, we introduce causal mediation by incorporating mediator $Z$, as depicted in Fig. 1 c. Derived from the image $X$, $Z$ serves as a differentiable proxy for $Y$, mediating the relationship to enhance performance. [2]

**Proposition 1. (Mediation in Causal Prediction Model):** Given a causal diagram $X \rightarrow Y$, introducing a mediator $Z$ to create the structure $X \rightarrow Z \rightarrow Y$, and assuming a strong correlation between $Y$ and $Z$, this results in: i) Conditional Independence: $(X \perp$

---

2. We do not explicitly include the site/scanner variable $S$ because the figure highlights only the variables directly used by the model ($X$, $Z$, $Y$, and the unobserved $U$). In practice, $S$ is an observed nuisance factor that influences $X$ (i.e., $S \rightarrow X$), but it does not enter the forward pass or the loss explicitly. $\mathcal{L}_{fd}$ and $\mathcal{L}_{\mathbf{fd}}^{\mathbf{exch}}$ therefore address site-driven variation indirectly (reducing the portion of appearance variability in $Z$ that is attributable to differences in $S$) without requiring site labels.

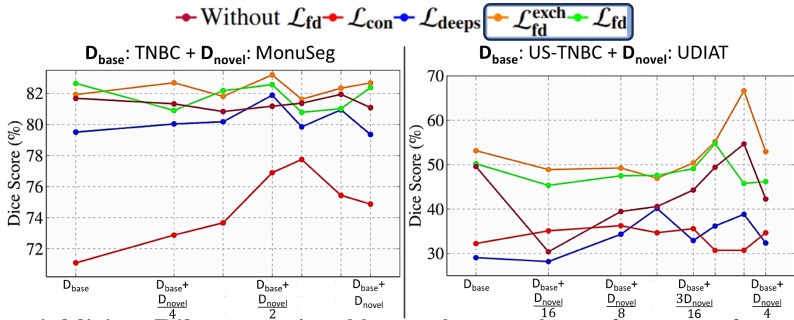

Figure 3: **Data Addition Dilemma.** An ablation showing the performance of various loss functions for histopathology and ultrasound datasets under the "Data Addition Dilemma" (Shen et al., 2024). Data from $D_{novel}$ is added to $D_{base}$ to observe how losses handle distribution shifts. $\mathcal{L}_{\mathbf{fd}}^{\mathbf{exch}}$ (orange) outperforms others in mitigating distribution shift when pooling data from multiple sources. The US-TNBC dataset has fewer samples, so we only added UDIAT dataset samples until their number matched (see Table 6.)

$Y$) | $Z$ ii) Preserved Modularity: $P(X) \perp P(Y \mid X)$ iii) Functional Relationship: $P(Y \mid X) = \int P(Y \mid Z)P(Z \mid X)$(Nazarovs et al., 2021)

The relationship shown in Proposition 1 indicates that $P(Y \mid X)$ depends on $P(Z \mid X)$, as $Z$ mediates $X \to Y$. This indicates that an accurate determination of $P(Z \mid X)$ allows for precise estimation of $P(Y \mid X)$.

**Example 1** *Consider $X \sim \mathcal{N}(0,1)$, where $\mathcal{N}$ denotes the normal distribution. Define $Z = aX + \epsilon_1$ and $Y = bZ + \epsilon_2$, where $\epsilon_1 \sim \mathcal{N}(0,1)$ and $\epsilon_2 \sim \mathcal{N}(0,1)$, and $a$ and $b$ are constants. Under these definitions, we have the following conditional distributions: $Z \mid X \sim \mathcal{N}(aX,1)$, $Y \mid Z \sim \mathcal{N}(bZ,1)$, and consequently $Y \mid X \sim \mathcal{N}(abX,1+b^2)$.*

The example demonstrates that $P(Y \mid X)$ is a function of $P(Z \mid X)$, as the mean of $Y \mid X$ (represented as $abX$) depends on the mean of $Z \mid X$ (which is $aX$). Moreover, conditional independence is preserved, as knowing $X$ provides no further information about $Y$ given $Z$ (Nazarovs et al., 2021).

## 2.2. Mediator as a Feature Discrepancy Measure

We now focus on identifying the appropriate mediator variable $Z$. From the previous section, we know that a good mediator $Z$ should exhibit a strong correlation with $Y$ and be derived from $X$. $Z$ corresponds to the intermediate representations of the U-Net architecture (Ronneberger et al., 2015), satisfying $X \to Z$. To ensure a strong correlation between $Y$ and $Z$, the foreground features of $Z$ must be highly discriminative from the background. In the U-Net architecture, the feature map $F$ is represented by height, width, and channel. Ground-truth masks or clustering methods during training help identify indicators $\tilde{y}$ that distinguish foreground from background features (Sims et al., 2023). To enforce this discriminative property, we penalize the following loss.

**Definition 1. (Feature Discrepancy Loss):** Let $F$ denote the features extracted from any network architecture and $\tilde{y}$ represent the indicator variables identifying foreground features (the ground truth segmentation mask). We define the channel-averaged foreground features as $F_g = \sum_k \left( \sum_{i,j} F[i,j,k] \otimes \tilde{y}[i,j,k] \right)$ and the channel-averaged background features as $B_g = \sum_{i,j} F[i,j,k] \otimes (1 - \tilde{y}[i,j,k])$, where $\otimes$ denotes element-wise multiplication.

The feature discrepancy loss is then given by:

$$\mathcal{L}_{\mathbf{fd}} = -\log\left(\|F_g - B_g\|^2\right) \tag{1}$$

In the previous discussion, $F_g - B_g$ represents the feature distance (FD) between foreground and background. Penalizing this feature difference helps the model distinguish between foreground and background better, reducing over- and under-segmentation. In Lemma 1, we prove that the negative logarithm of the Dice score lower bounds the feature-distance loss, indicating that minimizing the feature-distance loss can improve Dice scores (Supplementary Sec C.1 for proof).

**Lemma 1.** Relationship between feature discrepancy loss $\mathcal{L}_{\mathbf{fd}}$, segmentation Dice score, and constant $k$ for feature vector $F$ derived from image $X$:

$$-log(Dice \times (k+1)) \leq \mathcal{L}_{\mathbf{fd}} \tag{2}$$

An increase in the Dice score results in a decrease of the lower bound, which allows for a decrease in $\mathcal{L}_{\mathbf{fd}}$. As shown in Figure 1 (a), this relationship justifies the observed correlation between $\mathcal{L}_{\mathbf{fd}}$ and the Dice score for all models[3].

**Axiom 2** *(SGD Suboptimality for Convex Lipschitz Functions):* Let $f : \mathbb{R}^d \to \mathbb{R}$ *be a convex function with Lipschitz constant $L$. For step size $\eta_t = \frac{1}{L\sqrt{T}}$ and $T$ iterations, SGD achieves:* $\mathbb{E}\left[f(\theta_T) - f(\theta^*)\right] \leq \frac{CL}{\sqrt{T}}$, *where $C > 0$ is a constant, $\theta_T$ is the parameter at iteration $T$, and $\theta^*$ is the optimal parameter.*

Axiom 2 recalls a well-established fact that the suboptimality of Stochastic Gradient Descent (SGD) is bound by the Lipschitz constant ($L$) (Hardt et al., 2016). Details of the proof of Axiom 2 can be found in (Shalev-Shwartz and Ben-David, 2014) (page 187). In Lemma 2.2, we prove that minimizing $\mathcal{L}_{\mathbf{fd}}$ constrains the weight matrix $W$ in each UNet layer by damping gradient updates (Supplementary Sec C.2 for proof). This prevents $W$ from growing excessively to maximize feature discrepancy, which is crucial for medical imaging datasets that are significantly smaller than natural scene datasets. Large $W$ risks data interpolation and memorization contribute to high variance, increasing the risk of overfitting. By enforcing feature separation *without* relying on a large $||W||_2$, $\mathcal{L}_{\mathbf{fd}}$ acts as an implicit regularizer, effectively bounding The Lipschitz constant $L$ and improving generalization effect (see Axiom 2) empirically validated by higher test Dice scores (see Table 1). $\gamma$ controls the separation–norm trade-off via optimization. $\mathcal{L}_{\mathbf{fd}}$ balances overfitting and discrimination.

**Lemma 2. (Weight Norm Bound via Feature Discrepancy Loss):** Let $W \in \mathbb{R}^{d \times d}$ denote the weight matrix of a UNet layer producing features $F = W \otimes x$, where $x \in \mathbb{R}^{d \times d}$ is the input to that layer. With $x_g$ and $x_b$ denoting foreground and background features,

$$\mathcal{L}_{\mathbf{fd}} = -\log\left(||W \otimes (x_g - x_b)||_2^2\right) \tag{3}$$

Minimizing $\mathcal{L}_{\mathbf{fd}}$ bounds $||W||_2$, reducing the layer's Lipschitz constant.

---

3. Although Lemma 1's bound may not be tight, experiments (Figure 2 (b) and Table 1) show a strict upper-lower bound relationship, indicating that minimizing $\mathcal{L}_{\mathbf{fd}}$ directly improves the Dice score.

### 2.2.1. Implementation details

**Segmentation Loss $\mathcal{L}_{\mathbf{seg}}$.** To penalize spatial prediction, $\mathcal{L}_{\mathbf{seg}}$ integrates Dice loss (Soomro et al., 2018) and Binary Cross Entropy (BCE) loss (Jadon, 2020), both essential for image segmentation. These losses evaluate model performance by comparing expected and actual masks. $\mathcal{L}_{\mathbf{seg}}$ is a linear combination of Dice and BCE loss, as given in (Roy et al., 2024c) (Supplementary Sec B for more details).

**Layer-wise Feature Discrepancy Loss $\mathcal{L}_{\mathbf{fd}}$ and hyper-parameter $\alpha$ regulation.** The U-Net architecture employs an encoder-decoder structure with skip connections, enabling the extraction of multi-scale features at varying spatial resolutions. We introduce $\mathcal{L}_{\mathrm{fd}}$ at each feature layer to enhance segmentation accuracy. It is to be observed that the feature dimension for each layer of UNet is different. For applying $\mathcal{L}_{\mathbf{fd}}$ to each layer, we max pool $\tilde{y}$ to match the feature dimension of that particular layer to extract $F_g$ and $B_g$. This approach strengthens the model's discriminative power by encouraging distinct feature learning across layers, as illustrated in Fig. 1(b). A trainable hyperparameter $\alpha$ is introduced to control the importance of each layer in the feature discrepancy loss $\mathcal{L}_{\mathbf{fd}}$, with unique $\alpha$ values assigned per layer. This balances segmentation accuracy $\mathcal{L}_{\mathbf{seg}}$ and feature discrepancy at each level. Ablation (Section 3.3) reports the final $\alpha$ values, showing each layer's contribution to improved segmentation performance.

**Warm-Starting $\alpha$.** In the initial model updates, $\alpha = 0$, optimizing exclusively for $\mathcal{L}_{\mathbf{seg}}$ without factoring in the penalty function $\mathcal{L}_{\mathbf{fd}}$. This method enables $\alpha$ to progressively rise from zero to infinity, consistent with the literature (Bertsekas, 1997). This approach enables a seamless shift from a constrained to an unconstrained problem, allowing for a thorough exploration of the solution space. Starting with a small penalty helps to mitigate potential ill-conditioning associated with large penalties at the outset. We start with $\alpha$ set to 0, permitting the algorithm to iterate multiple times before activating $\alpha$ for training.

## 3. Experiments

We begin by detailing the experimental setup, including datasets, architectures, and a novel triple-negative breast cancer segmentation dataset. Sec 3.1 3.3 show how Feature Discrepancy Loss improves segmentation across datasets. In Sec 4, we demonstrate its effectiveness in preserving performance despite distributional shifts.

**Setup.** Experimental setup and dataset details can be seen in Supplementary Sec I. We assess three prominent U-Net variants: AttentionUNet (Jiménez et al., 2022), which employs gated attention mechanisms for improved segmentation; NucleiSegNet (Lal et al., 2021), designed for overlapping boundaries and varying nuclei sizes; and CMUNet (Tang et al., 2023), which combines multi-scale attention gates and a ConvMixer module to capture both global and local features (details in Supplementary Sec J).

### 3.1. Quantitative Results on individual datasets

The effects of $\mathcal{L}_{\mathbf{fd}}$ are detailed in Table 1, which presents results for all samples, as well as for the Worst-off and Best-off samples based on Dice scores. Table 6 presents the numbers of the best-off and worst-off samples utilized in our experiments. The worst-off samples are the samples with the lower dice scores than other samples (see Fig. 1 (a)) as seen without the application of $\mathcal{L}_{\mathbf{fd}}$. The best-off samples are the direct opposite of the worst-off samples,

| Model | Dataset | $\mathcal{L}_{fd}$ | Worst Off Samples | | | | Best Off Samples | | | | All Samples | | | |
|---|---|---|---|---|---|---|---|---|---|---|---|---|---|---|
| | | | Dice | Δ Dice | IoU | Δ IoU | Dice | Δ Dice | IoU | Δ IoU | Dice | Δ Dice | IoU | Δ IoU |
| AttnUNet (Jiménez et al., 2022) | UDIAT | ✗ | 22.42 | | 29.47 | | 75.86 | | 68.46 | | 67.21 | | 35.61 | |
| | | ✓ | **23.28** | +0.9 | **30.31** | +0.8 | **77.29** | +1.4 | **69.50** | +1.0 | **68.96** | +1.7 | **38.43** | +2.8 |
| | TNBC | ✗ | **77.88** | | 68.64 | | 85.82 | | 74.38 | | 80.61 | | 67.79 | |
| | | ✓ | 77.86 | 0.0 | **68.66** | +0.0 | **86.25** | +0.4 | **77.57** | +3.2 | **81.16** | +0.5 | **69.19** | +1.4 |
| | MoNuSeg | ✗ | 66.03 | | 52.38 | | 82.57 | | 73.48 | | 75.92 | | 61.28 | |
| | | ✓ | **68.61** | +2.5 | **53.06** | +0.7 | **83.62** | +1.0 | **74.50** | +1.0 | **77.97** | +2.0 | **62.87** | +1.6 |
| | AD 256 | ✗ | 56.35 | | 31.92 | | 81.34 | | 70.88 | | 61.14 | | 43.87 | |
| | | ✓ | **57.67** | +1.3 | **33.10** | +1.2 | **85.64** | +4.3 | **72.93** | +2.0 | **64.69** | +3.5 | **46.67** | +2.8 |
| CMUNet (Tang et al., 2023) | UDIAT | ✗ | 31.56 | | 26.58 | | 90.88 | | 88.25 | | 81.85 | | 69.87 | |
| | | ✓ | **33.19** | +1.6 | **28.17** | +1.6 | **95.32** | +4.4 | **90.01** | +1.8 | **84.22** | +2.4 | **73.02** | +3.1 |
| | US-TNBC | ✗ | 25.08 | | 21.44 | | 86.27 | | 68.09 | | 49.59 | | 34.53 | |
| | | ✓ | **26.94** | +1.9 | **22.35** | +0.9 | 86.04 | -0.2 | **69.35** | +1.3 | **50.22** | +0.6 | **36.52** | +2.0 |
| NuSegNet (Lal et al., 2021) | TNBC | ✗ | 77.29 | | 68.00 | | 86.49 | | 71.29 | | 81.69 | | 69.22 | |
| | | ✓ | **79.40** | +2.1 | **68.42** | +0.4 | **88.82** | +0.3 | **72.58** | +1.3 | **82.65** | +1.0 | **70.58** | +1.4 |
| | MoNuSeg | ✗ | 63.95 | | 50.05 | | 84.61 | | 70.40 | | 80.95 | | 67.91 | |
| | | ✓ | **64.61** | +0.7 | **52.11** | +2.1 | **84.96** | +0.3 | **71.65** | +1.2 | **81.69** | +0.7 | **68.65** | +0.7 |
| | AD 256 | ✗ | 32.55 | | 23.19 | | 64.75 | | 46.28 | | 51.15 | | 36.17 | |
| | | ✓ | **35.78** | +3.2 | **25.46** | +2.3 | **71.15** | +6.4 | **51.35** | +5.1 | **56.57** | +5.4 | **40.61** | +4.4 |

Table 1: **Ablation study on the application of $\mathcal{L}_{fd}$.** The improvement for low dice (Worst Off), high dice (Best Off), and all test samples (All Samples) is evident after applying $\mathcal{L}_{fd}$. NucleiSegNet (Jiménez et al., 2022) (histopathology) is not applicable to UDIAT and US-TNBC, while CMUNet (Tang et al., 2023) (ultrasound) does not apply to TNBC. Attention UNet (Jiménez et al., 2022) performs poorly on US-TNBC (Dice: 12.96). Changes in Dice (Δ Dice) and IoU (Δ IoU) are shown across all test settings.

and their count is equal to that of the worst-off samples. The threshold in Table 6 is an estimate of the approximate value of Dice scores below which the selected worst-off samples lie. In the case of CMUNet on the US-TNBC dataset, a slight decrease in the Dice score (-0.23) for Best-off samples is offset by improvements in Worst-off samples as $\mathcal{L}_{fd}$ increases the overall average performance while emphasizing the worst-off samples. On the new US-TNBC dataset, $\mathcal{L}_{fd}$ results in higher overall Dice scores. The improvements corroborate the theoretical findings in Lemma 1. (*Takeaway:* Penalizing $\mathcal{L}_{fd}$ enhances segmentation performance across models and datasets.)

Furthermore, we also demonstrate that the proposed $\mathcal{L}_{fd}$ is *not limited to binary segmentation* and naturally extends to multi-class segmentation settings (see Supplementary Sec L).

### 3.2. Qualitative Results on individual datasets

Qualitative results for the TNBC, MoNuSeg, AD, US-TNBC, and UDIAT datasets are presented in Figures 6. The red-highlighted areas in the predicted masks without $\mathcal{L}_{fd}$ indicate segmentation errors, while the green-highlighted regions reflect corrections made by applying $\mathcal{L}_{fd}$. These experiments illustrate how $\mathcal{L}_{fd}$ enhances segmentation through boundary refinement and reducing segmentation errors. The resulting masks display sharper, more accurate contours of key structures, preserving fine details and ensuring better anatomical representation. From Fig. 8, we can see that the introduction of $\mathcal{L}_{fd}$ significantly reduces the unnecessary activations and streamlines the focus of the model to the region of interest, thus enhancing the segmentation performance. (*Takeaway:* Penalizing for $\mathcal{L}_{fd}$ results in sharper boundaries, improved detail preservation, and increased consistency.)

### 3.3. Ablation Studies on Feature Discrepancy Loss and Dataset Performance

**Impact of the $\alpha$ Parameter on Feature Discrepancy Loss.** As discussed in Section 2.2.1, $\alpha$ is a trainable parameter that initially starts at zero and regulates the penalty of feature discrepancy loss, $\mathcal{L}_{\mathbf{fd}}$, for each layer of the neural network; the final values of $\alpha$ indicate that the layer with the highest value had the most influence on improving the overall dice scores (see Figure 2 (a)). Furthermore, applying $\mathcal{L}_{\mathbf{fd}}$ across all layers yielded consistently better Dice scores ($+1.3$–$1.8\%$ across datasets) compared to selective layers (Enc 1, Dec 4, Bot), indicating that refined features from earlier layers enhance the discriminative quality of the final segmentation output.

**Comparison with State-of-the-Art Models.** For the TNBC (Naylor et al., 2018), UDIAT (Yap et al., 2017), and MoNuSeg (Kumar et al., 2019) datasets, our method achieves Dice score improvements of $+0.96$ (TNBC), $+0.74$ (MoNuSeg), and $+0.75$ (UDIAT) compared to CMUNet (Tang et al., 2023) and NucleiSegNet (Lal et al., 2021), demonstrating the effectiveness of penalizing feature discrepancy in modalities with high foreground-background similarity. For AD, larger patches ($256 \times 256$ pixels) capture broader context, including background and neighboring pixels, while smaller patches ($128 \times 128$ pixels) focus primarily on plaque regions with limited context. Using the same experimental setup as (Jiménez et al., 2022), we observe performance improvements in AttnUNet (Jiménez et al., 2022) and NucleiSegNet (Lal et al., 2021) with $\mathcal{L}_{\mathbf{fd}}$, as shown in Fig. 7.

**Changes in $\mathcal{L}_{\mathbf{fd}}$ and Dice scores at the sample level.** In Figure 2 (a), a trend between $\mathcal{L}_{\mathbf{fd}}$ and Dice is noted, with some samples exhibiting poor scores in both metrics. Figure 2 (b) presents a frequency plot for $\mathcal{L}_{\mathbf{fd}}$ (orange) and Dice (blue). A shift in $\mathcal{L}_{\mathbf{fd}}$ to lower values and Dice scores to higher values is observed, indicating a significant improvement in Dice scores at the sample level. This can also be seen in Supplementary Sec D that the samples move towards a lower $\mathcal{L}_{\mathbf{fd}}$ and a higher Dice score region in the $\mathcal{L}_{\mathbf{fd}}$ vs Dice plot. We also validate that the experiments are statistically significant in Supplementary Sec G.

**Performance comparison of various loss functions under noisy data.** Supplementary Sec K highlights the robustness of $\mathcal{L}_{\mathbf{fd}}$ over other loss functions due to the disentanglement between the foreground-background features, ensuring robust discriminative features under noisy data conditions.

## 4. Mitigating Data Distribution Shifts Under Assumed Exchangeability

Recent work emphasizes expanding medical imaging datasets by pooling data from multiple sources (Chytas et al., 2024; Lokhande et al., 2022; Roy et al., 2026). While early efforts apply invariant representation learning to handle covariate shifts, they often address only limited factors(Akash et al., 2021). The **Data Addition Dilemma** (Shen et al., 2024), underscores a critical issue: increasing training data size across sources can induce distributional shifts that degrade model performance. Traditional methods based on independent and identically distributed (i.i.d.) assumptions fail in cross-dataset scenarios. *Why i.i.d. is not realistic?* While the i.i.d. assumption is standard in most machine learning pipelines and often effective, it becomes overly restrictive in data addition scenarios. Exchangeability, being a weaker and more realistic assumption, better reflects the practical data generation process. For instance, in curating datasets like US-TNBC, new samples often depend on previously collected batches, violating i.i.d. but remaining consistent with exchangeability. We formalize exchangeability as a foundational assumption (Axiom 3 Supplementary

Sec. C.3) ensuring that the joint distribution of multi-source data remains invariant under permutation. This weaker assumption provides a theoretically sound basis for pooling without introducing bias from distributional shifts, and is consistent with the causal mediation framework in Section 2.1.

This challenge arises when combining a novel dataset, $\mathcal{D}_{\mathbf{novel}}$, with a base dataset, $\mathcal{D}_{\mathbf{base}}$, as their joint use violates i.i.d. assumptions. To address this, we leverage **exchangeability**, which extends beyond i.i.d. by ensuring that the joint distribution remains invariant under index permutations (Definition 2). By treating $\mathcal{D}_{\mathbf{base}}$ and $\mathcal{D}_{\mathbf{novel}}$ as exchangeable, we design a modified penalty loss function spanning both datasets. This ensures that discrepancies between foreground and background features across datasets remain comparable to within-dataset discrepancies, mitigating distributional shifts effectively. Algorithm 1 (Supplementary Sec E) outlines the training process using $\mathcal{L}_{\mathbf{fd}}^{\mathbf{exch}}$.

**Definition 2. (Feature Discrepancy Loss under assumed exchangeability):** $F_g(\mathcal{D})/$ $B_g(\mathcal{D})$ represents foreground/background features from a randomly sampled dataset $\mathcal{D}$, which can be either $\mathcal{D}_{\mathbf{novel}}$ or $\mathcal{D}_{\mathbf{base}}$ dataset.

$$\mathcal{L}_{\mathbf{fd}}^{\mathbf{exch}} = -\log\left(\|F_g(\mathcal{D}_{\mathbf{base}}) - B_g(\mathcal{D}_{\mathbf{novel}})\|^2 + \|F_g(\mathcal{D}_{\mathbf{novel}}) - B_g(\mathcal{D}_{\mathbf{base}})\|^2\right) \qquad (4)$$

### 4.1. Experiments on Data Addition Dilemma

We selected TNBC as our base dataset, denoted as $\mathcal{D}_{\mathbf{base}}$, using the MoNuSeg dataset as our novel dataset, labeled $\mathcal{D}_{\mathbf{novel}}$. We added samples from MoNuSeg sequentially, to $\mathcal{D}_{\mathbf{base}}$ (For example, in the first setup we use $D_{base} + \frac{D_{novel}}{16}$ while testing it on $D_{base}$, in the next setup we use $D_{base} + \frac{D_{novel}}{8}$ while testing it on $D_{base}$ and so on). All evaluations were performed on $\mathcal{D}_{\mathbf{base}}$. Similarly, for the ultrasound datasets, we designated US-TNBC as $\mathcal{D}_{\mathbf{base}}$ and UDIAT as $\mathcal{D}_{\mathbf{novel}}$, with samples from UDIAT added in batches of 15 images. We compared three methods: a naive method without penalties, a method penalizing for $\mathcal{L}_{\mathbf{fd}}$, and a method penalizing for $\mathcal{L}_{\mathbf{fd}} + \mathcal{L}_{\mathbf{fd}}^{\mathbf{exch}}$. We compare these three losses with the existing losses that deal with disentanglement ($\mathcal{L}_{\mathbf{con}}$(Chaitanya et al., 2020)) and layer-wise supervision ($\mathcal{L}_{\mathbf{deeps}}$(Dou et al., 2016)). Notably, the naive method, $\mathcal{L}_{\mathbf{con}}$(Chaitanya et al., 2020) and $\mathcal{L}_{\mathbf{deeps}}$(Dou et al., 2016) exhibited a decrease in test set accuracy on $\mathcal{D}_{\mathbf{base}}$ as more samples from $\mathcal{D}_{\mathbf{novel}}$ were incorporated, consistent with the findings of (Shen et al., 2024). $\mathcal{L}_{\mathbf{fd}} + \mathcal{L}_{\mathbf{fd}}^{\mathbf{exch}}$ resulted in an overall performance improvement, as illustrated in Fig. 3. While $\mathcal{L}_{\mathbf{fd}}^{\mathbf{exch}}$ shares conceptual similarity with local/pixel-wise contrastive losses (Chaitanya et al., 2020), Fig. 3 shows contrastive loss ($L_{con}$) suffers significant performance drops (7–19%). This aligns with prior findings (e.g., Sec. 2 of (Akash et al., 2021)) showing contrastive losses require complex modifications to handle data addition and distribution shifts due to strong i.i.d. assumptions, making the weaker exchangeability assumption more realistic.

To rigorously explain the non-monotonic trends observed in Fig. 3, we quantify the distributional mismatch introduced at the data-addition stages. We compute the Kullback–Leibler (KL) divergence and Jensen–Shannon (JS) distance between $\mathcal{D}_{\mathbf{base}}$ and the incrementally added subset of $\mathcal{D}_{\mathbf{novel}}$. Images are converted to grayscale using the luminosity method, and pixel intensity distributions are estimated via normalized histograms. Metrics are computed separately for: (i) foreground pixels (mask $= 1$), (ii) background pixels (mask $= 0$), and (iii) the overall image distribution. A small $\epsilon$ is added to avoid numerical instability in KL computation.

Table 2: Distributional divergence between TNBC ($\mathcal{D}_{\mathbf{base}}$) and MoNuSeg ($\mathcal{D}_{\mathbf{novel}}$). FG and BG represent Foreground and Background respectively.

| Addition Step | FG KL | FG JS | BG KL | BG JS | Overall KL | Overall JS |
|---|---|---|---|---|---|---|
| $\mathcal{D}_{\mathbf{base}} + \mathcal{D}_{\mathbf{novel}}/4$ | 0.6534 | 0.3390 | 0.7653 | 0.4034 | 0.7249 | 0.3844 |
| $\mathcal{D}_{\mathbf{base}} + \mathcal{D}_{\mathbf{novel}}/2$ | 0.2125 | 0.1948 | 0.3301 | 0.2516 | 0.2995 | 0.2471 |
| $\mathcal{D}_{\mathbf{base}} + \mathcal{D}_{\mathbf{novel}}$ | 0.2227 | 0.2173 | 0.3501 | 0.2792 | 0.3160 | 0.2622 |

*i) Table 2 and Fig. 3:* As we can see, the distributional difference between $\mathcal{D}_{\mathbf{base}}$ and $\mathcal{D}_{\mathbf{novel}}/4$ is high and this indicates lower performance as seen in all baselines. With increase in more data, the distributional difference between $\mathcal{D}_{\mathbf{base}}$ and $\mathcal{D}_{\mathbf{novel}}/2$ also reduces leading to peak in performance. Furthermore, the distributional difference between $\mathcal{D}_{\mathbf{base}}$ and $\mathcal{D}_{\mathbf{novel}}$ increases which correlates with an overall dip in the performance of the baselines in that region. This explain the overall trend seen in Fig. 3.

Table 3: Distributional divergence between US-TNBC ($\mathcal{D}_{\mathbf{base}}$) and UDIAT ($\mathcal{D}_{\mathbf{novel}}$). FG and BG represent Foreground and Background respectively.

| Addition Step | FG KL | FG JS | BG KL | BG JS | Overall KL | Overall JS |
|---|---|---|---|---|---|---|
| $\mathcal{D}_{\mathbf{base}} + \mathcal{D}_{\mathbf{novel}}/16$ | 0.0502 | 0.1083 | 0.0225 | 0.0612 | 0.0247 | 0.0662 |
| $\mathcal{D}_{\mathbf{base}} + \mathcal{D}_{\mathbf{novel}}/8$ | 0.0502 | 0.1083 | 0.0225 | 0.0612 | 0.0247 | 0.0662 |
| $\mathcal{D}_{\mathbf{base}} + 3\mathcal{D}_{\mathbf{novel}}/16$ | 0.0110 | 0.0509 | 0.0019 | 0.0218 | 0.0022 | 0.0234 |
| $\mathcal{D}_{\mathbf{base}} + \mathcal{D}_{\mathbf{novel}}/4$ | 0.0166 | 0.0628 | 0.0041 | 0.0310 | 0.0043 | 0.0318 |

*ii) Table 3 and Fig. 3:* Similar to Table 2, the distributional difference between $\mathcal{D}_{\mathbf{base}}$ and $\mathcal{D}_{\mathbf{novel}}/16$ is high and this indicates lower performance as seen in all baselines. The distributional difference remains same for $\mathcal{D}_{\mathbf{base}}$ and $\mathcal{D}_{\mathbf{novel}}/8$ while the slight increment in performance seen in some baselines are indication of more availability of data. he distributional difference between $\mathcal{D}_{\mathbf{base}}$ and $3\mathcal{D}_{\mathbf{novel}}/16$, and $\mathcal{D}_{\mathbf{base}}$ and $\mathcal{D}_{\mathbf{novel}}/4$ is significantly less than the previous data addition regions, showing improved performance of the baselines (and thus the peaks). $\mathcal{L}_{con}$ is the worst performing baseline for both the data addition setups as contrastive objectives inherently rely on data for performance and is unable to show trends or maintain comparable performance with respect to other baselines due to the data scarcity in medical imaging.

## 5. Conclusion

Data scarcity is a major challenge in medical imaging. To address this, our research introduces a novel feature discrepancy penalty function ($\mathcal{L}_{fd}$) that enhances segmentation performance across modalities like histopathology and ultrasound. Our method outperforms existing models and baselines, showing improved Dice scores for both the worst-off and best-off samples. To tackle the lack of datasets for triple-negative breast cancer (TNBC), we introduced a new ultrasound dataset focused on TNBC. $\mathcal{L}_{fd}$ reduces erroneous activation maps, enabling models to focus on relevant spatial regions more effectively. This is particularly impactful in the "Data Addition Dilemma" scenario, where pooling data from multiple sources introduces distribution shifts that degrade model performance. A modified version, $\mathcal{L}_{fd}^{excg}$, incorporates feature exchangeability to mitigate these shifts.

## Acknowledgments

Prof. Lokhande acknowledges support from University at Buffalo startup funds, an Adobe Research Gift, an NVIDIA Academic Grant, and the National Center for Advancing Translational Sciences of the NIH (award UM1TR005296 to the University at Buffalo). Prof. Kim acknowledges support from RS-2019-II191906 (Graduate School of AI at POSTECH).

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

## Appendix A. Related Works

Recent advances in deep learning have surpassed traditional segmentation methods like watersheds (Beucher and Meyer, 2018) and super-pixels (Li and Chen, 2015), demonstrating high efficacy in medical image segmentation (Long et al., 2015; Ronneberger et al., 2015; Chen et al., 2014; Singha and Bhowmik, 2023; Keaton et al., 2023; Kanadath et al., 2023). While MCFNet (Feng et al., 2021) captures spatial information, it struggles with complex staining patterns. Multimodal approaches (Dwivedi et al., 2022; Roy et al., 2024b; Chen et al., 2021b; Tomar et al., 2022a; Zhao et al., 2024b; Roy et al., 2024a) integrate spatial and textual data but face challenges with homogeneous pixel distributions in medical images.

Loss-based approaches like contrastive loss (Chaitanya et al., 2020; Xu et al., 2021), deep supervision (Dou et al., 2016), and entropy minimization (Fleuret et al., 2021) improve

U-Net (Ronneberger et al., 2015) segmentation by refining foreground and background representations. Entropy minimization and contrastive losses separate pixel-level class representations. Our method also uses foreground-background feature discrepancy but penalizes all U-Net layers. Deep supervision applies binary cross-entropy loss to decoder outputs without contrastive losses. Approaches like (He et al., 2021; Gong et al., 2024) enhance class consistency and feature re-ranking, but none penalize feature discrepancy across all layers. Our work is the first to show that layer-wise foreground-background discrepancy improves U-Net (Ronneberger et al., 2015) representations, addressing the "data addition dilemma" (Shen et al., 2024). In addition to that, all the aforementioned losses are based on the strong i.i.d. assumption, which does not hold true always, as discussed in Sections 1 and 5. *Furthermore, segmentation tasks involve assigning a label to each pixel (or region) in an image, resulting in a structured output with strong spatial dependencies among pixels or regions, unlike classification that assigns a single label to the entire image (treating each sample as an independent entity).* In segmentation, the data points (pixels or regions) within an image are not independent; their labels are often correlated due to spatial continuity and object boundaries. Therefore, the i.i.d. assumption is often violated, making exchangeability, a weaker and more realistic assumption, more appropriate.

SAM (Kirillov et al., 2023), though powerful, is unsuitable for medical images due to its reliance on prompts and inability to handle numerous objects of interest without guidance (Mazurowski et al., 2023). Large models like Transformers require extensive data, which is scarce in medical imaging (He et al., 2023), and fine-tuning pre-trained models can introduce modality biases (Barnett et al., 2024; Kumar et al., 2022) and unobserved task-irrelevant confounders (U in Fig. 1)(Zhang et al., 2024). Furthermore, Transformer architectures like TransUNet (Chen et al., 2021a) and SwinUNet (Cao et al., 2022) suffer from oversegmentation and are unable to determine the foreground structure accurately compared to their UNet counterparts (Zhao et al., 2024a) (see Supplementary Sec F) for a comparison of the transformer architectures with our method). These challenges explain why U-Net variants remain the most widely adopted segmentation models in medical imaging (Yousef et al., 2023; Ahmadi et al., 2023). [4]

## Appendix B. The segmentation loss

The Dice loss (Soomro et al., 2018) and Binary Cross Entropy (BCE) loss (Jadon, 2020) are crucial for image segmentation tasks, evaluating model performance by comparing predicted and actual masks. The dice loss ($L_{dice}$) and the BCE loss ($L_{bce}$) are defined in Eq. 5 and 6 respectively where $y_{ijk}$ represents the ground truth label for pixel $(i, j, k)$, $\tilde{y}_{ijk}$ represents the predicted probability for pixel $(i, j, k)$, $\epsilon$ is a small constant added for numerical stability to avoid division by zero or taking the log of zero, and $N$ is the total number of elements

---

4. We clarify that approaches like SIFA (Chen et al., 2019) were not included in the current study because *they address a different problem setting.* Methods like SIFA explicitly assume a source-target paradigm with known domain labels and rely on adversarial feature alignment to match global distributions. In contrast, our work focuses on data-scarce pooling scenarios where datasets are incrementally added, domain boundaries may be ambiguous, and no source-target distinction or domain labels are assumed. *Forcing such methods into our setting would require introducing artificial source-target splits and domain labels, fundamentally altering the problem formulation and leading to an unfair comparison.*

pixels.

$$L_{dice} = 1 - \frac{2\sum_{i,j,k} y_{ijk} \cdot \tilde{y}_{ijk} + \epsilon}{\sum_{i,j,k} y_{ijk} + \sum_{i,j,k} \tilde{y}_{ijk} + \epsilon} \tag{5}$$

$$L_{bce} = -\frac{1}{N} \sum_{i,j,k} \Big( y_{ijk} \cdot \log(\tilde{y}_{ijk}) \\ + (1 - y_{ijk}) \cdot \log(1 - \tilde{y}_{ijk}) + \epsilon\Big) \tag{6}$$

We use a linear combination of $L_{dice}$ and $L_{bce}$ as $L_{seg}$ (Roy et al., 2024c). This can be seen in Eq. 7

$$L_{seg} = L_{dice} + L_{bce} \tag{7}$$

## Appendix C. Proofs

### C.1. Lemma 1

**Lemma 1.** Relationship between feature discrepancy loss $\mathcal{L}_{\mathbf{fd}}$, segmentation Dice score, and constant $k$ for feature vector $F$ derived from image $X$:

$$-log(Dice \times (k+1)) \leq \mathcal{L}_{\mathbf{fd}} \tag{8}$$

**Proof** Let $\otimes$ denote element-wise multiplication. Then we get the relation between Dice score, the predicted segmentation mask $y$, and the ground truth segmentation mask $\tilde{y}$ as:

$$\sum_{i,j,k} \tilde{y}_{ijk} = \frac{Dice}{2} \times \frac{\sum_{i,j,k} y_{ijk} + \sum_{i,j,k} \tilde{y}_{ijk}}{\sum_{i,j,k} y_{ijk}} \\ (\text{since } Dice = \frac{2\sum_{i,j,k} y_{ijk} \cdot \tilde{y}_{ijk} + \epsilon}{\sum_{i,j,k} y_{ijk} + \sum_{i,j,k} \tilde{y}_{ijk} + \epsilon}) \tag{9}$$

Now, simplifying FD (feature discrepancy) using Definition 1, we get:

$$FD = \frac{\| \sum_k \left( \sum_{i,j} F_{i,j,k} \otimes \tilde{y}_{i,j,k} - \sum_{i,j} F_{i,j,k} \otimes (1 - \tilde{y}_{i,j,k}) \right) \|_2}{\| \sum_{i,j,k} F_{ijk} \|_2} \tag{10}$$

$$FD \leq \frac{\| 2\sum_{i,j,k} F_{i,j,k} \otimes \tilde{y}_{i,j,k} \|_2}{\| \sum_{i,j,k} F_{ijk} \|_2} + \frac{\| \sum_{i,j,k} F_{i,j,k} \|_2}{\| \sum_{i,j,k} F_{ijk} \|_2} \\ (\text{using the triangle inequality}). \tag{11}$$

Now, using Eq. 9 we get:

$$FD - 1 \leq \frac{\| \sum_{i,j,k} F_{i,j,k} \otimes Dice \times \frac{\sum_{i,j,k} y_{ijk} + \sum_{i,j,k} \hat{y}_{ijk}}{\sum_{i,j,k} y_{ijk}} \|_2}{\| \sum_{i,j,k} F_{ijk} \|_2} \tag{12}$$

Since $\sum_{i,j,k} \tilde{y}_{ijk}$ and $\sum_{i,j,k} y_{ijk}$ are constants during testing, let $\frac{\sum_{i,j,k} \tilde{y}_{ijk}}{\sum_{i,j,k} y_{ijk}} = k'$:

$$-log(FD) \geq -log(Dice \times (k+1))$$
$$\text{(Taking} -\log \text{ on both sides).} \tag{13}$$

$$\mathcal{L}_{\mathbf{fd}} \geq -log(Dice \times (k+1)) \tag{14}$$

This completes the proof. ∎

### C.2. Lemma 2

**Lemma 2. (Weight Norm Bound via Feature Discrepancy Loss):** Let $W \in \mathbb{R}^{d \times d}$ denote the weight matrix of a UNet layer producing features $F = W \otimes x$, where $x \in \mathbb{R}^{d \times d}$ is the input to that layer. The relationship between $\mathcal{L}_{\mathbf{fd}}$ and $W$ is given by:

$$\mathcal{L}_{\mathrm{fd}} = -\log\left(||W \otimes (x_g - x_b)||_2^2\right) \tag{15}$$

where $x_g$ and $x_b$ are foreground and background features of $x$, respectively. Minimizing $\mathcal{L}_{\mathrm{fd}}$ implicitly enforces an upper bound on the spectral norm $||W||_2$, reducing the layer's Lipschitz constant and improving generalization.

**Proof** Let $\Delta x = x_g - x_b$ denote the inherent foreground-background separation in the input space. The loss $\mathcal{L}_{\mathrm{fd}}$ incentivizes maximizing $||W \otimes \Delta x||_2^2$ where $\otimes$ is the hadamard product. Now we can frame the hadamard product in a different way to represent $W \otimes \Delta x$ as $W_{exp} \times \Delta x_{exp}$ where $\times$ is matrix multiplication, $W_{exp} \in \mathbb{R}^{d^2 \times d^2}$ is a diagonalized form of $W$ and $x_{exp} \in \mathbb{R}^{d^2 \times 1}$ is a reshaped form of $x$. The Lipschitz constant $L$ of the layer $F = W_{exp} \times x$ is the spectral norm of $W_{exp}$:

$$L = ||W_{exp}||_2 = \sup_{||x_{exp}||_2 = 1} ||W_{exp} \times x_{exp}||_2 \tag{16}$$

This measures the maximum amplification of the input by $W_{exp}$. To minimize $\mathcal{L}_{\mathrm{fd}}$, the optimization ensures $||W_{exp} \times \Delta x_{exp}||_2^2 \geq \gamma$ for some $\gamma > 0$. By Cauchy-Schwarz:

$$||W_{exp} \times \Delta x_{exp}||_2 \leq ||W_{exp}||_2 ||\Delta x_{exp}||_2. \tag{17}$$

Squaring both sides:

$$\gamma \leq ||W_{exp} \times \Delta x_{exp}||_2^2 \leq ||W_{exp}||_2^2 ||\Delta x_{exp}||_2^2 \tag{18}$$

$$\implies ||W_{exp}||_2 \geq \frac{\sqrt{\gamma}}{||\Delta x_{exp}||_2}. \tag{19}$$

Thus, $\gamma$ defines the *minimum required separation* between foreground and background features.

The gradient of $\mathcal{L}_{\text{fd}}$ with respect to $W$ is:

$$\nabla_W \mathcal{L}_{\text{fd}} = -\frac{2}{||W \otimes \Delta x||_2^2}(W \otimes \Delta x)(\Delta x)^T. \tag{20}$$

The term $\frac{1}{||W \otimes \Delta x||_2^2}$ acts as an *adaptive damping factor*: as $||W \otimes \Delta x||_2^2$ increases (better separation), the gradient magnitude decreases. This prevents $W$ from growing excessively to inflate separation artificially, thereby bounding $||W||_2$ and thus $||W_{exp}||_2$ (since $W_{exp}$ is a diagonalized form of $W$).

The network achieves $||W_{exp} \times \Delta x_{exp}||_2^2 \geq \gamma$ with the smallest possible $||W||_2$ (due to gradient damping) ensuring *lower variance model* (reduced sensitivity to input perturbations) and preventing overfitting. Furthermore, the Lipschitz constant $L$, is also reduced, indicating a *tighter generalization bounds* (the suboptimal error bound, $\mathcal{E}_{\text{gen}} \propto L$ as seen in Axiom 2). ∎

### C.3. Axiom 3

**Axiom 3 (Exchangeability of Pooled Data)** *Let $\{(X_i, Y_i, S_i)\}_{i=1}^n$ denote a collection of data points from multiple sources, where $S_i$ indicates the source (e.g., scanner or site). The sequence is* exchangeable *if for any permutation $\pi$ of the indices $\{1, \ldots, n\}$, the joint distribution satisfies:*

$$P((X_1, Y_1, S_1), \ldots, (X_n, Y_n, S_n)) = P((X_{\pi(1)}, Y_{\pi(1)}, S_{\pi(1)}), \ldots, (X_{\pi(n)}, Y_{\pi(n)}, S_{\pi(n)})). \tag{21}$$

*This implies that the order of samples carries no information about their joint distribution, even when $S_i$ induces dependence.*

**Proof Sketch.** Exchangeability is a standard assumption in Bayesian and causal inference (e.g., de Finetti's theorem). In our setting, it is justified by the causal graph in Fig. 2c the confounders $U$ influence both $X$ and $Y$, but are conditionally independent of the source $S$ given the mediator $Z$. Under the front-door adjustment via $Z$, the causal effect $X \to Y$ is identifiable and invariant across sources. Consequently, permuting the source labels does not change the joint distribution of $(X, Y, Z)$, satisfying exchangeability. This provides the theoretical foundation for the exchangeable feature discrepancy loss $\mathcal{L}_{\text{fd}}^{\text{exch}}$ (Definition 2), which treats foreground/background features from $\mathcal{D}_{\text{base}}$ and $\mathcal{D}_{\text{novel}}$ as interchangeable, thereby mitigating distributional shifts.

## Appendix D. Qualitative analysis of Dice vs Feature discrepancy loss

Figure 4 shows the improvement in the Dice scores of the samples with a decrease in $\mathcal{L}_{\textbf{fd}}$. As we can see in the Dice vs $\mathcal{L}_{\textbf{fd}}$ plot, as Dice score improves, $\mathcal{L}_{\textbf{fd}}$ decreases, and the points move to the top left corner of the plot. The green arrows signify the movement of each sample (each test image) after the use of $\mathcal{L}_{\textbf{fd}}$. The red arrow indicates an overall

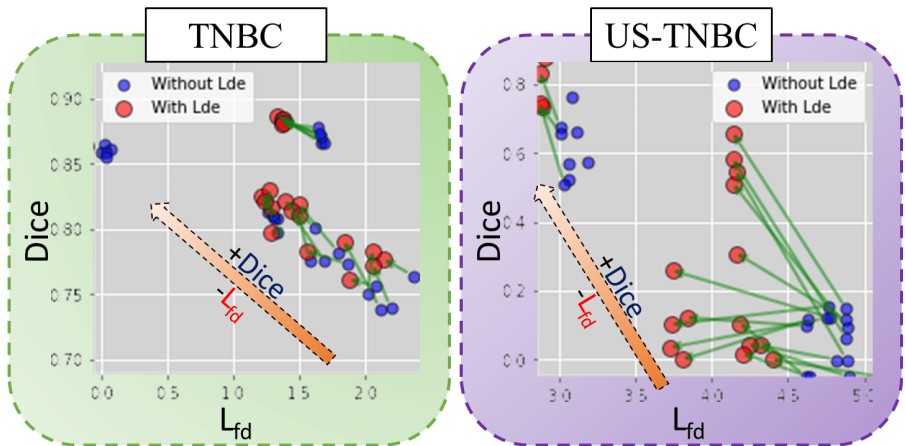

Figure 4: **The change in Dice scores with change in $\mathcal{L}_{\mathbf{fd}}$.** The plot with axis as Dice score and $\mathcal{L}_{\mathbf{fd}}$ for samples of TNBC (Naylor et al., 2018) and US-TNBC for the Bottleneck (Bot) layer of NucleiSegNet (Lal et al., 2021) and CMUNet (Tang et al., 2023) are plotted respectively. The green arrows indicate the movement of each point after the use of $\mathcal{L}_{\mathbf{fd}}$. The red arrow indicates the overall movement of the majority of the samples.

movement of the samples highlighting the flow towards the top left corner (increase in Dice and decrease in $\mathcal{L}_{\mathbf{fd}}$). This signifies the importance of foreground and background feature disentanglement to ensure robust medical image segmentation under a data-scarce setting with complex backgrounds.

## Appendix E. Algorithmic explanation of exchangeable Feature discrepancy loss

The algorithmic explanation of $\mathcal{L}_{\mathbf{fd}}^{\mathbf{exch}}$ for each iteration of training can be seen in Algorithm 1. Specifically, the foreground feature of image $i$ $(F_{g,i})$ pushes the background feature of image $j$ $(B_{g,j})$ in $\mathcal{L}_{\mathbf{fd}}^{\mathbf{exch}}$, while $F_{g,j}$ simultaneously pushes $B_{g,j}$ in $\mathcal{L}_{\mathbf{fd}}$. This draws $F_{g,i}$ and $F_{g,j}$ closer, minimizing the distributional shift caused by differences in batch data sources.

---
**Algorithm 1** $\mathcal{L}_{\mathbf{fd}}^{\mathbf{exch}}$ explained in Sec 4.1
---
**Input:** Foreground features $F_g$ and background features $B_g$ for each image in a batch of size $n$
**for** *each training iteration* **do**
    **for** $i = 1$ **to** $n$ **do**
        $\mathcal{L}_{\mathrm{fd}} = -\log(\|F_{g,i} - B_{g,i}\|_2)$
        $\mathcal{L}_{\mathbf{fd}}^{\mathbf{exch}} = -\log(\|F_{g,i} - B_{g,i+k}\|_2)$
        $L_i = \mathcal{L}_{\mathrm{fd}} + \mathcal{L}_{\mathbf{fd}}^{\mathbf{exch}}$
    **end**
    loss $\leftarrow \frac{1}{n}\sum_{i=1}^{n} \alpha L_i$
**end**
**return** *loss*

---

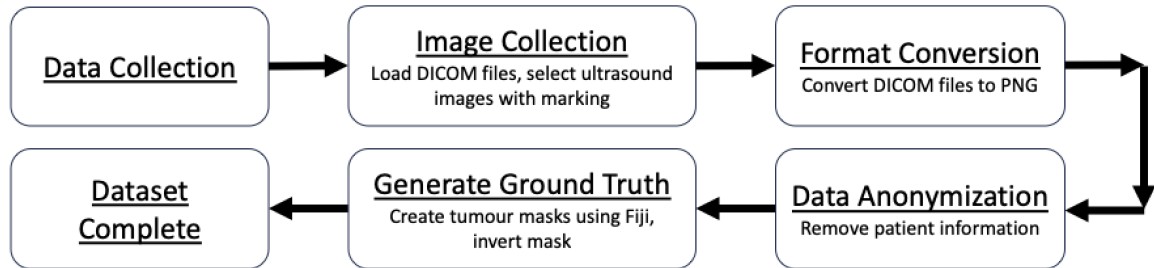

Figure 5: The steps involved in the creation of the US-TNBC dataset.

## Appendix F. Comparison with Transformer architectures

Table 4: Comparison with Transformer architectures. Scores are in %.

| Model | TNBC | | MonuSeg | |
|---|---|---|---|---|
| | Dice | IoU | Dice | IoU |
| TransUNet (Chen et al., 2021a) | 76.50 | 72.04 | 77.46 | 63.85 |
| SwinUNet (Cao et al., 2022) | 60.08 | 50.14 | 76.38 | 62.54 |
| Ours | 82.65 | 70.58 | 81.69 | 68.65 |

As seen in Table 4, the transformer architectures are data hungry models and are not suitable for data-scarce medical image segmentation tasks. We see that TransUnet and SwinUnet have lower Dice and IoU scores even with respect to the base UNet models (See Table 2). This shows that data-hungry Transformer architectures are not an ideal choice for medical domains with data scarcity.

## Appendix G. Significance testing

To rigorously evaluate the impact of the $L_{fd}$, we conducted a statistical significance test. The baseline model (for example, CMUNet without $L_{fd}$) and five runs of the baseline model with $L_{fd}$ was used for a one-sample t-test. It can be seen from Table 5 that $p$-values<0.01, which makes the experiments statistically significant.

## Appendix H. Details about the US-TNBC dataset

The TNBC dataset focuses on Triple-Negative Breast Cancer tissues. The images are typically 721 x 570 pixels in size on average. It consists of 30 images, including 15 ultrasound images and 15 ground truth images. The data collected at baseline includes breast ultrasound images of women aged between 42 and 76 years old. This data was collected between 2022 and 2023, and the images are in PNG format. To make the acquired data useful, some refinement tasks were performed. Firstly, the DICOM images were loaded into a DICOM reader, and the tumor images without marking or annotation were selected. Next, the DICOM files were converted into PNG format. The patient information was also eliminated using image cropping software. The images were cropped to retain maximum anatomical

| MoNuSeg | | | US-TNBC | | |
|---|---|---|---|---|---|
| Model | Dice | IoU | Model | Dice | IoU |
| NuSegNet | 0.0015 | 0.0032 | CMUNet | 0.0045 | 0.0051 |
| AttnUNet | 0.0019 | 0.0007 | | | |

Table 5: Statistical significance ($p$-values).

| Dataset | Data Type | All Samples | Worst Off | Threshold |
|---|---|---|---|---|
| TNBC (Naylor et al., 2018) | Histopathology | 50 | 10 | 75.0 |
| MoNuSeg (Kumar et al., 2019) | Histopathology | 44 | 25 | 70.0 |
| UDIAT (Yap et al., 2017) | Ultrasound | 163 | 35 | 25.5 |
| US-TNBC | Ultrasound | 15 | 10 | 65.5 |
| AD (Jiménez et al., 2022) | Histopathology | 10k | 500 | 40.0 |

Table 6: **Summary of datasets.** "All Samples" denote all the test samples of the dataset, whereas "Worst Off" are the test samples with lower Dice scores. Threshold is an approximate estimate of the range of Dice score below which the "Worst Off" samples lie.

information while removing unnecessary boundaries and markers. The ground truth images were generated using Fiji, an open-source image processing program based on ImageJ2. The ground truth masks were produced and then inverted to match the UDIAT dataset mask convention, where the tumor masks are white and the background is black. This dataset is designed to evaluate algorithms for cancer detection, grading, and classification. The steps involved in the collection of the US-TNBC dataset are shown in Fig. 9. The dataset request link will be made accessible along with the code.

## Appendix I. Experimental Setup

We conduct experiments using four datasets: the TNBC dataset (Naylor et al., 2018) with histopathology images featuring dense glandular tissues and indistinct boundaries; the MonuSeg dataset (Kumar et al., 2019), which includes Hematoxylin and Eosin-stained histopathology images; and a novel US-TNBC dataset comprising 15 ultrasound images of TNBC tissues collected in 2022-23, with ground truth masks generated using Fiji (more details in Supplementary Sec H). The UDIAT dataset (Yap et al., 2017) includes breast ultrasound images characterized by irregular tumor morphology and indistinct boundaries. Additionally, we evaluate an Alzheimer's histopathology dataset (Jiménez et al., 2022) for tau protein segmentation, with AD $256 \times 256$ and AD $128 \times 128$ versions, where the former contains a more complex background. Causal mediation and control of $\mathcal{L}_{\mathbf{fd}}$ are independent of the neural network architecture.

## Appendix J. Implementation details

We developed our segmentation model using Python and implemented it with the Tensor-Flow and Keras libraries. For data processing, we utilized numpy, OpenCV, and scikit-learn, enabling efficient data handling. We have used the high-performance NVIDIA TESLA P100 GPU to accelerate training and leverage hardware acceleration. The model has been trained for 100 epochs in the initial phase ($\alpha = 0$) and 75 epochs in the second phase with $\mathcal{L}_{\mathrm{fd}}$

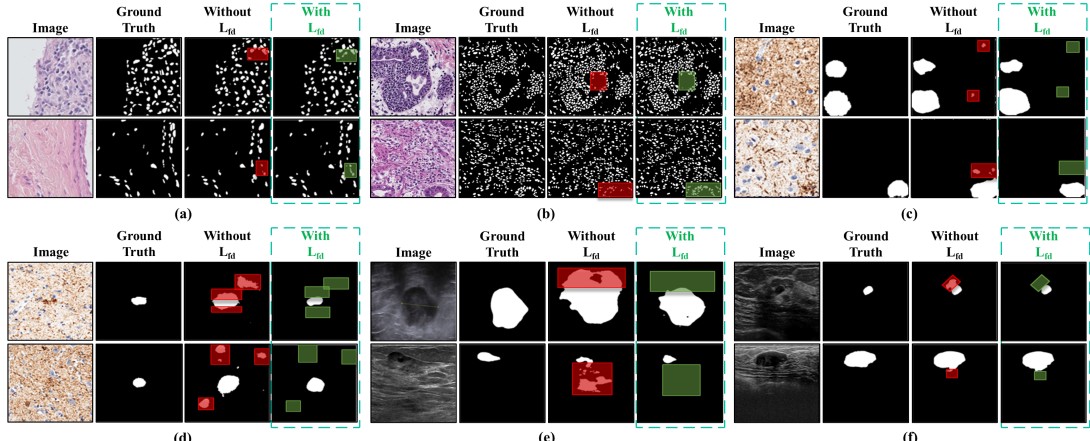

Figure 6: **Qualitative analysis of histopathology and ultrasound segmentation.** Results for (a) TNBC, (b) MoNuSeg, (c) AD 128, and (d) AD 256 datasets of NucleiSegNet, and for (e) US-TNBC and (f) UDIAT datasets of CMUNet, with (green) and without $\mathcal{L}_{\mathbf{fd}}$. Red boxes highlight faulty segmentation without $\mathcal{L}_{\mathbf{fd}}$, while green boxes show improvements with $\mathcal{L}_{\mathbf{fd}}$.

($\alpha \neq 0$). It has been seen that this specific initialization of $\alpha$ produces the best results as compared to other initialization values. This can be intuitively explained as the model learns to produce decent segmentation masks using the traditional $L_{seg}$ and a regularization by $\mathcal{L}_{\text{fd}}$ utilizes the prior knowledge gained by the model in the 75 epochs to refine the feature maps. [5] This effectively achieves a better Dice score (+1.5-1.9% across all datasets) than training the model with $\mathcal{L}_{\text{fd}}$ from the beginning. A train-test-validation split of 70-20-10% has been applied. Callbacks were used to save the best-performing model during both training phases. To address non-uniform image sizes, all images have been resized to uniform $512 \times 512$ pixels for TNBC (Naylor et al., 2018), the newly collected US-TNBC, and $256 \times 256$ for UDIAT (Yap et al., 2017) and AD (Jiménez et al., 2022) (both $256 \times 256$ and $128 \times 128$). We have applied data augmentation (horizontal and vertical flipping, rotations to the left and right by 90°) on the training set to train the models and on the test set to increase the number of data points for the plots. Evaluation of the models has been done on the test set without augmentation. Further details will be available with the code.

## Appendix K. Performance comparison

Fig. 7 illustrates the performance of the state-of-the art methods comapred to ours. It is evident that the application of $\mathcal{L}_{fd}$ provides the ecessary boost to surpass the performance of the current methods. This also translates to the qualitative analysis as seen in Fig. 7. Heatmaps seen in Fig. 8 indicate the refinement in the focus of the model's convolution

---

5. *We set $\alpha = 75$ to allow the network to first learn stable and semantically meaningful foreground–background priors from the base data before introducing $\mathcal{L}_{fd}$. Empirically, $\alpha > 75$ does not yield further performance gains, indicating that these priors and the corresponding feature representations have already converged, and that delaying the regularizer further provides no additional benefit. In contrast, applying $\mathcal{L}_{fd}$ too early ($\alpha < 75$) leads to suboptimal performance, as foreground–background priors are not yet sufficiently formed; enforcing discrepancy at this stage acts on unstable features and introduces improper constraints. This behavior highlights the importance of activating $\mathcal{L}_{fd}$ only after reliable foreground–background priors have emerged.*

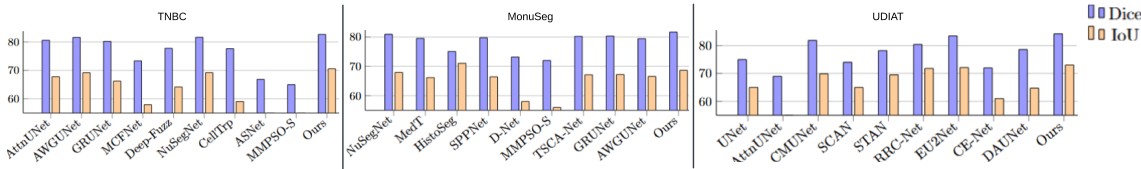

Figure 7: **Comparison of the proposed method with existing models for various datasets.** We compare our model with MedT (Valanarasu et al., 2021), HistoSeg (Wazir and Fraz, 2022), SPPNet (Xu et al., 2023), D-Net (Islam Sumon et al., 2023), MMPSO-S (Kanadath et al., 2023), TSCA-Net (Fu et al., 2024), GRUNet (Roy et al., 2024a) and AWGUNet (Roy et al., 2024b) for MoNuSeg (Kumar et al., 2019), AWGUNet (Roy et al., 2024b), GRUNet (Roy et al., 2024a), MCFNet (Feng et al., 2021), Deep-Fuzz (Das et al., 2023), CellTrp (Keaton et al., 2023), ASNet (Singha and Bhowmik, 2023), MMPSO-S (Kanadath et al., 2023) for TNBC(Naylor et al., 2018), and UNet (Ronneberger et al., 2015), SCAN (Zhang et al., 2020), STAN (Shareef et al., 2020), RRC-Net (Chen et al., 2023), $EU^2Net$ (Roy et al., 2024c), CE-Net (Gu et al., 2019), and DAUNet (Pramanik et al., 2024) for UDIAT(Yap et al., 2017). For AD 128 and 256, we compare our proposed method with UNet (Jiménez et al., 2022), AttnUNet (Jiménez et al., 2022), and NuSegNet (Tomar et al., 2022b). Our proposed loss improves the Dice score of the best-performing architecture for AD 128 (NuSegNet (Tomar et al., 2022b)) and AD 256 (AttnUNet (Jiménez et al., 2022)) by 1.74 and 3.55, respectively.

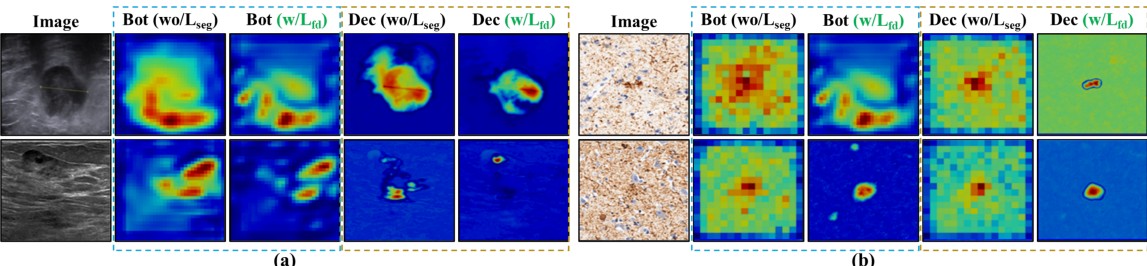

Figure 8: **Heatmaps with and without the use of $\mathcal{L}_{fd}$.** Heatmaps of the bottleneck layer (Bot) and the last decoder layer (Dec) of NucleiSegNet and CMUNet for AD and US-TNBC respectively. With $\mathcal{L}_{fd}$ (green), the erroneous activation maps are reduced, leading to better prediction.

layers after the application of $\mathcal{L}_{\mathrm{fd}}$ leads to a finer predicted segmentation mask with lesser non overlapping regions corresponding to the ground truth.

To evaluate model robustness under noisy conditions, Gaussian noise was systematically added to the images. For each image $I \in \mathbb{R}^{H \times W \times C}$ in the dataset, zero-mean Gaussian noise $\epsilon$ with standard deviation $\sigma$ was sampled:

$$\epsilon \sim \mathcal{N}(0, \sigma^2) \tag{22}$$

The noisy image $\tilde{I}$ was then computed as:

$$\tilde{I} = \mathrm{clip}(I + \epsilon, 0, 1) \tag{23}$$

where $\sigma$ was varied across experiments ($\sigma \in \{0.05, 0.10, 0.15, 0.20\}$) and clip($\cdot$) ensured valid pixel intensities. This process maintained original data dimensions while simulating realistic sensor noise artifacts. In this experiment, we see that the proposed $\mathcal{L}_{\mathbf{fd}}$ has lesser dip in the performance as compared to $L_{con}$, $L_{deeps}$ and $L_{seg}$ (Without $\mathcal{L}_{\mathbf{fd}}$, i.e., a combination of Dice loss and BCE loss). This indicates that foreground-background feature disentanglement ensures robust feature extraction even for noisy/poor-quality images.

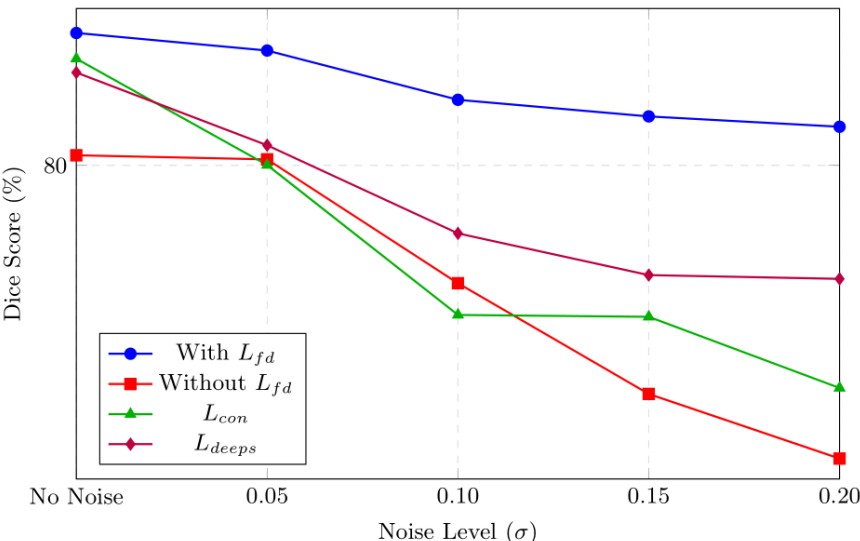

Figure 9: Analysis of the loss functions in the presence of noise added to input images while training. Dice score dips are less for $\mathcal{L}_{\mathbf{fd}}$ as compared to other losses as the strength of the noise increases.

## Appendix L. Extension to Multi-class settings

To demonstrate the effectiveness of $\mathcal{L}_{fd}$, we evaluate our method on the CoNSeP dataset (Graham et al., 2019), a standard benchmark for *multi-class nuclei instance segmentation*, which contains multiple semantic nucleus categories annotated at the instance level. Following common practice in nuclei segmentation, the multi-class problem is decomposed into multiple binary foreground–background segmentation tasks, one per nucleus class (e.g., inflammatory, epithelial, spindle-shaped, etc.). Each binary task predicts a class-specific foreground mask against a shared background, and the final multi-class segmentation is obtained by aggregating class-wise predictions. This strategy is widely adopted in prior work, as it allows class-specific feature learning while preserving a consistent background representation. $\mathcal{L}_{fd}$ operates at the level of *foreground vs. background feature separation* and does not assume a single foreground category. In the multi-class setting, $\mathcal{L}_{fd}$ is applied independently to each class-specific binary segmentation head, encouraging robust and disentangled foreground–background representations for every nucleus type. As a result, the method scales linearly with the number of classes and does not require any modification to the loss formulation. All experiments use a standard U-Net architecture following the nnU-Net training configuration, with no architectural changes. We report class-wise and overall performance on the CoNSeP (Graham et al., 2019) validation/test split using standard nuclei segmentation metrics. Class-wise metrics are computed independently for each nucleus category, and overall performance is obtained by averaging across classes. Table 7 confirm that $\mathcal{L}_f d$ is not restricted to binary segmentation, but applies directly to multi-class tasks through standard binary decomposition, while retaining its benefits in disentangling

Table 7: Multi-class nuclei segmentation results (Dice score) on CoNSeP.

| Method | Background | Other | Inflammatory | Healthy Epithelial | Malignant Epithelial |
|---|---|---|---|---|---|
| UNet | 91.94 | 0.87 | 37.03 | 16.37 | 58.67 |
| UNet+$\mathcal{L}_{fd}$ | 91.77 | 1.82 | 34.59 | 32.93 | 59.53 |

| Method | Fibroblast | Muscle | Endothelial |
|---|---|---|---|
| UNet | 18.73 | 25.41 | 0.00 |
| UNet+$\mathcal{L}_{fd}$ | 28.18 | 36.48 | 0.00 |

foreground–background representations under distribution shift. We will include this results in the supplementary of the camera-ready version.

