# OpenReview forum: "Is Exchangeability better than I.I.D to handle Data Distribution Shifts while Pooling Data for Data-scarce Medical image segmentation?"
_MIDL.io/2026/Conference — MIDL 2026 Poster_

### Official Review · Reviewer_yuu2 · 2026-01-06

**Confidence:** 3
**Preliminary Rating:** 3
**Final Rating:** 3

**Summary:**

This paper investigates medical image segmentation under data-scarce pooling and distribution shifts, arguing that treating datasets as exchangeable (rather than strictly i.i.d.) better reflects real-world multi-source training. The authors propose a foreground–background feature discrepancy regularizer applied across multiple layers of U-Net–style models, and an `exchangeable variant that couples foreground/background feature statistics between a base dataset and an added novel dataset to mitigate the data addition dilemma.

**Strengths:**

- The paper addresses a common real-world issue in medical segmentation: pooling data from different sources can hurt performance due to distribution shifts, especially in data-scarce settings.
- The proposed foreground–background feature discrepancy loss is lightweight, easy to implement, and can be added to existing U-Net–style models without architectural changes.

**Weaknesses:**

- The `exchangeability' framing is not tightly connected to the method. The proposed approach is essentially a cross-dataset foreground/background feature regularizer; it is unclear what specific, testable implication of exchangeability is being used, or why this framing is necessary beyond standard domain-shift/robustness arguments.
- Novelty relative to existing segmentation regularizers is not clearly distinguished. Encouraging separability between foreground and background features is conceptually close to prior ideas such as deep supervision and feature-level separation objectives; the paper should more clearly state what is new (e.g., specific formulation, layer-wise application, exchangeable coupling) and why simpler alternatives are insufficient.
- The experimental improvements are often small, and results are mostly reported as single numbers without multi-seed variance or confidence intervals, making it hard to judge whether gains are consistent and statistically meaningful.

**Detailed Comments:**

- Tighten the discussion around exchangeability/causality: either formalize the assumptions and their implications more precisely, or reduce over-strong statements and present the method as a domain-shift regularizer.
- Please clarify the exact computation of the feature discrepancy loss at each layer (how masks are downsampled/aligned to feature maps, how foreground/background regions are pooled, and whether any normalization is applied).
- For baseline fairness, please state whether competing regularizers/losses were tuned with comparable effort, and include at least a short sensitivity analysis for key loss weights.

**Justification Of Final Rating:**

Thanks for the detailed response. The rebuttal improves the connection between the “exchangeability” motivation and the proposed cross-dataset FG/BG feature-statistics constraint, which helps clarify the intended mechanism. However, the current evidence for robustness would be stronger with explicit multi-seed mean±std/CI in the main results (beyond significance tests), and the practicality under limited/noisy labels in the novel dataset remains unclear since the loss relies on ground-truth masks from both domains. Based on these considerations, I would still keep my rating as borderline.

**Justification Of The Preliminary Rating:**

I currently give a Borderline rating. The problem setting is important and the proposed regularizer is simple and potentially useful for mitigating performance drops under cross-dataset pooling. However, my score is limited by (i) the unclear connection between the exchangeability/causal framing and the actual loss formulation, and (ii) the modest reported gains without variance-aware reporting (multiple seeds / confidence intervals), which makes it hard to assess robustness. If the rebuttal can clearly justify the exchangeability claim, document fair tuning protocols, and provide stability results across seeds (or equivalent evidence), I would lean toward Weak Accept.

**Questions To Address In The Rebuttal:**

- What concrete, testable property of exchangeability is leveraged by the proposed loss, and how does this differ from standard domain-shift regularization or feature alignment?
- Are the reported improvements stable across multiple random seeds and split variations?
- How were hyperparameters (loss weights, sampling ratios) selected for both the proposed method and competing regularizers? A brief description of the tuning protocol or a sensitivity analysis would address comparability concerns.

---

> ### Author Response · Authors · 2026-01-24
>
> We thank the reviewer for noting that the *“proposed foreground–background feature discrepancy loss is lightweight, easy to implement, and can be added to existing U-Net–style models without architectural changes.”* We also appreciate the recognition of the problem statement and its relevance to medical imaging.
>
> ---
>
> ### **@ Testable Property of Exchangeability Leveraged by the Proposed Loss**
>
> #### **i) Reiterating the Data Addition Dilemma**
>
> Consider two histopathology datasets ( D_A ) and ( D_B ) collected at different sites with distinct staining and background textures. When training begins on ( D_A ), the network learns foreground–background feature representations that partially entangle semantic content (nuclei vs. background) with dataset-specific appearance cues (e.g., staining intensity, scanner noise).
>
> When a small batch from ( D_B ) is added, performance may temporarily improve if the new data aligns with these priors. However, as more samples from ( D_B ) are introduced, the mismatch between priors learned from ( D_A ) and the appearance statistics of ( D_B ) leads to unstable optimization and performance degradation. This order- and proportion-dependence of performance is the **Data Addition Dilemma**.
>
> ---
>
> #### **ii) Role of Exchangeability — Intuition → Testable Constraint**
>
> The core issue is that standard training implicitly learns a **dataset-dependent prior** over foreground–background features. Let
> $Z_f = \text{foreground features}, \quad Z_b = \text{background features}$.
> Without constraints, the learned distributions
> $P(Z_f \mid D_A), ; P(Z_b \mid D_A)$
> become tied to appearance statistics of ( D_A ). When ( D_B ) is added, its features are interpreted through priors shaped by ( D_A ), causing representation mismatch and instability.
>
> ---
>
> #### **Exchangeability Assumption (What is relaxed)**
>
> We do **not** assume identical image distributions:
> $P(X \mid D_A) \neq P(X \mid D_B)$.
>
> Instead, exchangeability assumes only that the **semantic roles** of foreground and background are invariant across datasets:
> $P(Z_f \mid \text{FG}, D_A) \approx P(Z_f \mid \text{FG}, D_B), \quad
> P(Z_b \mid \text{BG}, D_A) \approx P(Z_b \mid \text{BG}, D_B).$
>
> This is weaker than i.i.d. and matches medical pooling scenarios: staining and noise differ, but “nucleus vs. background” meaning does not change.
>
> ---
>
> #### **Testable Property Enforced by ( $\mathcal{L}_{fd}^{\text{exch}}$ )**
>
> The above assumption yields a **concrete, testable constraint**:
>
> > **Foreground–background feature statistics should be invariant to dataset identity.**
>
> Operationally, ( $\mathcal{L}_{fd}^{\text{exch}}$ ) penalizes discrepancies between foreground and background feature statistics computed separately on ( D_A ) and ( D_B ). Thus, it enforces:
> $| \mu_f^{A} - \mu_f^{B} | + | \mu_b^{A} - \mu_b^{B} | \rightarrow 0$
> (and similarly for higher-order statistics).
>
> This directly prevents the model from encoding dataset identity into foreground–background structure. As new data is added, priors cannot drift toward appearance-specific cues; instead, the representation must remain **exchangeable** across datasets.
>
> ---
>
> #### **How This Mitigates the Data Addition Dilemma**
>
> The dilemma arises from **prior mismatch** when new data conflicts with priors learned early. By enforcing exchangeable feature structure:
>
> * Priors over ( Z_f, Z_b ) are constrained to be **dataset-invariant**
> * Representations depend less on which dataset appears first
> * Performance becomes less sensitive to data order and proportion
>
> This stabilization under incremental pooling is empirically observed in **Figure 3**, where ( $\mathcal{L}_{fd}^{\text{exch}}$ ) consistently outperforms standard segmentation losses under distribution shift (also detailed in our response on *Unclear Trends in Data Addition* to Reviewer 6spk).
>
> ---
>
> ### **@ Stability of the Reported Results**
>
> The stability of improvements across random seeds is supported by the statistical significance analysis in **Table 3** and **Appendix G**.
>
> ---
>
> ### **@ Hyperparameter Selection (Loss Weights, Sampling Ratios)**
>
> All methods were tuned using the **same validation protocol on the base dataset**.
>
> * Loss weights were chosen from a small predefined grid and fixed across experiments.
> * Competing regularizers were tuned using the same budget and validation splits, either following original papers or equivalent grid searches.
>
> We will clarify this in **Appendix J** of the revised/camera-ready version.
>
> ---
>
> The reviewer has acknowledged the practical relevance, methodological simplicity, and applicability of our approach. Given the clarifications and additional empirical analysis provided, we respectfully request reconsideration of the score.

---

> > ### Author Response · Authors · 2026-01-29
> >
> > Dear reviewer,
> > we have added additional experimentations and justifications to the concerns raised and are eager to engage in a discussion to further resolve any other confusion or concern that needs to be addressed in the paper. Looking forward to your response.

---

### Official Review · Reviewer_6spk · 2026-01-09

**Confidence:** 3
**Preliminary Rating:** 4

**Summary:**

The authors address the "Data Addition Dilemma" in medical image segmentation, where pooling data from multiple sources (or adding novel datasets) can degrade model performance due to distribution shifts. To mitigate this, the paper proposes a "Feature Discrepancy Loss" $L_{fd}$ designed to maximize the distinction between foreground and background features across all layers of a U-Net architecture. Theoretical justifications are provided using a causal mediation framework. Furthermore, the authors introduce a variant of the loss, $L_{fd}^{exch}$, based on the assumption of exchangeability rather than the traditional i.i.d. assumption, to better handle multi-source data pooling. The method is validated on five datasets, including a newly curated ultrasound dataset (US-TNBC), showing state-of-the-art performance and robustness against noise.

**Strengths:**

Relevance: The paper targets a critical and often overlooked issue in medical imaging: the "Data Addition Dilemma". Addressing why more data sometimes hurts performance is of high value to the MIDL community.

Reproducibility: The code is available on GitHub.

Broad Empirical Evaluation: The method is evaluated across five distinct datasets covering both histopathology and ultrasound modalities. The consistent improvement over strong baselines demonstrates the robustness of the proposed approach.

Theoretical Motivation: The authors attempt to ground their loss function in learning theory. Specifically, Lemma 2, which links the minimization of $L_{fd}$​ to bounding the spectral norm of the weight matrix (and thus the Lipschitz constant), provides a nice theoretical intuition for why the method improves generalization and reduces overfitting.

Performance on "Worst-off" Samples: I appreciate the analysis specifically highlighting improvements in "worst-off" samples (Table 1). It shows that the method effectively handles difficult edge cases rather than just boosting the average slightly.

**Weaknesses:**

Causal Framework Justification: The application of the Front-Door Adjustment is not rigorously justified. The Front-Door criterion requires that the mediator Z isolates the causal path from X to Y and that there is no direct path from the confounder U to Z. However, in a neural network, Z (feature maps) is a deterministic function of the input X. If the confounder U (e.g., scanner artifacts) causes changes in X, then U inherently influences Z through X. Additionally, annotator shift is a major potential confounder. Different sites may define Y differently for similar X (or if there were overlap/repeated annotations). In this case, the annotator acts as an unobserved confounder that is not blocked by X. Since X (image appearance) does not fully encode the "Annotation Protocol," violating the conditions required for the mathematical proofs. The paper admits this is a separate challenge, but it should be acknowledged that this violates the causal assumptions.

Unclear Trends in Data Addition (Figure 3): In Figure 3 in the US-TNBC + UDIAT plot, there is a sharp performance peak at $D_{base}+{3/16 * D_{novel})$ followed by a significant drop at for the proposed method. The "Data Addition Dilemma" usually implies a monotonic degradation or a plateau, but such volatile peaks and valleys suggest potential instability or sensitivity to the specific batches added, which is not fully explained in the text.

Mathematical Terminology (Axioms vs. Theorems): The paper repeatedly refers to established mathematical results as "Axioms". For example, "Axiom 2" refers to the suboptimality of SGD for convex Lipschitz functions. This is a derived theorem/result, not an axiom. Similarly, "Axiom 3" describes the definition of exchangeability.

Mathematical Terminology (Axioms vs. Theorems): The paper repeatedly refers to established mathematical results as "Axioms". For example, "Axiom 2" refers to the suboptimality of SGD for convex Lipschitz functions. From my perspective this is a derived theorem/result, not an axiom. Similarly, "Axiom 3" describes the definition of exchangeability.

Hyperparameter Complexity: The method relies on a warm-start strategy (starting $\alpha=0$) and layer-wise learnable weights $\alpha$. The sensitivity of the model to the timing of this warm-start (e.g., why exactly 75 epochs for the second phase?) is not fully explored.

**Detailed Comments:**

Figure 1(a) Axis Labeling: The y-axis in Figure 1(a) (labeled "Dice") ranges from 0 to 2.5 or 2. Since Dice coefficients are mathematically bounded between 0 and 1, the axis should likely be labeled as "Dice (%)" or scaled to [0, 1] to avoid confusion. And if it is [0-100] why show only that range?

Missing Keywords: The keyword section currently reads "List of keywords, comma separated," the authors forgot to fill this in.

Mathematical Terminology: Please correct the usage of "Axiom." An axiom is a starting assumption that cannot be proven (e.g., Euclid's axioms). The SGD convergence bound (Axiom 2) is a Theorem or Lemma. The definition of Exchangeability (Axiom 3) is a Definition.

**Justification Of The Preliminary Rating:**

I lean towards a Weak Accept. The paper addresses a significant problem in the medical domain (the Data Addition Dilemma) and provides a practical, effective solution supported by extensive experiments and a new dataset contribution. The availability of code is a strong plus. However, there are notable weaknesses: the theoretical framing (using "Axioms" for theorems) is mathematically imprecise, and the causal justification is shaky. Furthermore, the unexplained volatility in the data addition experiments (Figure 3) raises questions about stability. If the authors can clarify the experimental trends and correct the mathematical terminology, the paper would be solid.

**Questions To Address In The Rebuttal:**

Figure 3 Trends: Can you explain the volatility in Figure 3 (right), specifically why performance peaks at $D_{base}+{3/16 * D_{novel})$ and then drops? Is this due to specific characteristics of the added data batch, and does it imply the method is sensitive to which data is added?

Annotator Variation: Did the pooled datasets (e.g., TNBC vs. MoNuSeg) use different annotation protocols? are there repeated images with multiple annotations?

Causal Justification: Can you clarify how the Front-Door criterion holds given that Z is deterministically computed from X, and X is caused by U? Does this not imply a path U->X->Z? How does your method formally block the influence of U on Z beyond standard regularization?

Terminology: Please address the usage of the term "Axiom" for derived theorems. Would you be willing to rephrase these as "Theorems" or "Lemmas"?

---

> ### Author Response · Authors · 2026-01-24
>
> We thank the reviewer for acknowledging *“The paper targets a critical and often overlooked issue in medical imaging: the Data Addition Dilemma and is of high value to the MIDL community.”* We are glad the reviewer found the method an *“effective solution supported by extensive experiments and a new dataset contribution.”*
>
> ---
>
> ### **@ Causal Framework Justification**
>
> From a causal perspective, nuisance factors U (e.g., scanner or staining variations) influence the image X, which is deterministically mapped to features Z, implying the path U → X → Z. We do not claim to block this path as in strict front-door adjustment; instead, we constrain how information from U is encoded in Z.
>
> Under standard training, Z entangles task-relevant structure (foreground vs. background) with dataset-specific appearance cues induced by U, making representations dependent on early dominant datasets and causing instability during incremental data addition.
>
> Our loss acts as a representation-level constraint, enforcing consistent foreground–background feature separation across datasets despite appearance differences. This biases the model toward encoding stable predictive structure while reducing reliance on spurious correlations from U.
>
> Although information about U may remain in Z, its effect on the foreground–background decision boundary is weakened. This level of disentanglement is sufficient to stabilize learning under distribution shift without domain labels, adversarial alignment, or assumptions required for full causal identifiability.
>
> ---
>
> ### **@ Unclear Trends in Data Addition (Figure 3)**
>
> To rigorously explain the non-monotonic trends observed in Figure 3, we quantify the **distributional mismatch** introduced at the data-addition stages. We compute **Kullback–Leibler (KL) divergence** and **Jensen–Shannon (JS) distance** between **D_base** and incrementally added subsets of **D_novel**. Images are converted to grayscale using the luminosity method, and pixel intensity distributions are estimated via normalized histograms. Metrics are computed separately for:
> 1. Foreground pixels (mask = 1)
> 2. Background pixels (mask = 0)
> 3. Overall image distribution
> A small **ε** is added to avoid numerical instability in KL computation.
>
> ---
>
> #### **Table 1 — TNBC (D_base) vs MoNuSeg (D_novel)**
>
> | Addition Step      | FG KL  | FG JS  | BG KL  | BG JS  | Overall KL | Overall JS |
> | ------------------ | ------ | ------ | ------ | ------ | ---------- | ---------- |
> | D_base + D_novel/4 | 0.6534 | 0.3390 | 0.7653 | 0.4034 | 0.7249     | 0.3844     |
> | D_base + D_novel/2 | 0.2125 | 0.1948 | 0.3301 | 0.2516 | 0.2995     | 0.2471     |
> | D_base + D_novel   | 0.2227 | 0.2173 | 0.3501 | 0.2792 | 0.3160     | 0.2622     |
>
> **Interpretation (Table 1 + Fig. 3):**
> The divergence between **D_base** and **D_novel/4** is high, correlating with performance drops across baselines. As more data is added (**D_novel/2**), divergence reduces substantially, corresponding to **performance peaks**. When the full dataset is added, divergence increases again, which aligns with the **subsequent performance dip**.
>
> ---
>
> #### **Table 2 — US-TNBC (D_base) vs UDIAT (D_novel)**
>
> | Addition Step        | FG KL  | FG JS  | BG KL  | BG JS  | Overall KL | Overall JS |
> | -------------------- | ------ | ------ | ------ | ------ | ---------- | ---------- |
> | D_base + D_novel/16  | 0.0502 | 0.1083 | 0.0225 | 0.0612 | 0.0247     | 0.0662     |
> | D_base + D_novel/8   | 0.0502 | 0.1083 | 0.0225 | 0.0612 | 0.0247     | 0.0662     |
> | D_base + 3D_novel/16 | 0.0110 | 0.0509 | 0.0019 | 0.0218 | 0.0022     | 0.0234     |
> | D_base + D_novel/4   | 0.0166 | 0.0628 | 0.0041 | 0.0310 | 0.0043     | 0.0318     |
>
> **Interpretation (Table 2 + Fig. 3):**
> Divergence is relatively high in early additions (**/16**, **/8**), explaining lower performance. Divergence drops significantly at **3D_novel/16** and **D_novel/4**, matching the **performance peaks**. The contrastive baseline performs worst because contrastive objectives inherently rely on abundant data, making them unstable in data-scarce medical regimes.
>
> ---
>
> ### **@ Mathematical Terminology**
>
> Great suggestion. We will rename: **Axiom 2 → Lemma** and **Axiom 3 → Definition**.
>
> ---
>
> ### **@ Hyperparameter Complexity**
>
> We set α = 75 so the network first learns stable foreground–background priors before introducing **$𝓛_{fd}$**. Empirically, α > 75 yields no gain (priors already converged), while α < 75 degrades performance because the regularizer acts on unstable early features. Thus, **$𝓛_{fd}$** should activate only after reliable priors emerge, ensuring it operates on meaningful representations.
>
> ---
>
> ### **@ Figure 1(a) Axis**
>
> Figure 1(a) shows Dice₁ + Dice₂ + Dice₃ from deep supervision at three decoder stages, not a single Dice score. Since each Dice ∈ [0,1], their sum approaches 3. The plot analyzes correlation between **$𝓛_{fd}$** and multi-scale segmentation quality rather than absolute Dice.
>
> ---

---

> > ### Author Response · Authors · 2026-01-29
> >
> > Dear reviewer,
> > we have added additional experimentations and justifications to the concerns raised and are eager to engage in a discussion to further resolve any other confusion or concern that needs to be addressed in the paper. Looking forward to your response.

---

### Official Review · Reviewer_Zv9N · 2026-01-10

**Confidence:** 4
**Preliminary Rating:** 2
**Final Rating:** 3

**Summary:**

The paper explores the better way to pool data for data-scarce medical image segmentation, under the Data Addition Dilemma. The paper introduces feature discrepancy loss under assumed exchangeability and demonstrates its effectiveness in handling training with multi-source data. The paper is well-presented.

**Strengths:**

1. The paper is well-motivated by a practical problem in the medical imaging segmentation field, and comprehensively studies Data Addition Dilemma.
2. The paper proposed feature discrepancy loss and it seems to effectively leverage multi-source data.
3. The paper is well-presented, with a professional format of the results.

**Weaknesses:**

1. The proposed feature discrepancy loss seems to only be feasible for binary segmentation tasks, as it assumes the pixels to be only background and foreground. If so, it would limit the contribution in medical imaging application, as many tasks are multi-class.
2. Multi-source data training has been explored and some key baselines are missing, for example, SIFA.
3. What would be performance if the two datasets are of large distribution shift, for example, one from CT and the other from MRI.

**Detailed Comments:**

See Strengths and Weaknesses

**Justification Of Final Rating:**

I really appreciate the authors' detailed response and additional clarifications and experiments on multi-class tasks. The results makes the paper more convincing and I am, therefore, happy to raise my score.

**Justification Of The Preliminary Rating:**

The paper comprehensively studies the Data Addition Dilemma and proposes feature discrepancy-related loss to mitigate the distribution shift. However, the loss seems to limit the application only in binary segmentation and more baselines should be included.

**Questions To Address In The Rebuttal:**

Add experiments on more modalities and include key baselines.

---

> ### Author Response · Authors · 2026-01-24
>
> We thank the reviewer for highlighting that *“The paper is well-motivated by a practical problem in the medical imaging segmentation field, and comprehensively studies Data Addition Dilemma”* and that *“the paper explores the better way to pool data for data-scarce medical image segmentation, under the Data Addition Dilemma”*. We further thank the reviewer for acknowledging that *“The paper is well-presented, with a professional format of the results.”*
>
> ---
>
> ### **@ Not applicable to multi-class tasks**
>
> We clarify that $\mathcal{L}_{fd}$ is not restricted to binary segmentation and extends naturally to multi-class settings. We validate this on CoNSeP, a benchmark for multi-class nuclei instance segmentation. As standard in nuclei literature, the task is decomposed into class-wise foreground–background binary segmentations, and final predictions are obtained by aggregating class outputs.
>
> Because $\mathcal{L}{fd}$ enforces foreground–background feature separation, it applies independently to each class head without modifying the loss formulation, scaling linearly with the number of classes. All experiments use a standard U-Net (nnU-Net configuration). We report class-wise and mean Dice on CoNSeP, confirming that $\mathcal{L}{fd}$ remains effective in multi-class segmentation. Results are shown in Table 1 and will be included in the supplementary.
>
> #### **Table 1 — Multi-class nuclei segmentation results (Dice score) on CoNSeP**
>
> | Method                    | Background | Other | Inflammatory | Healthy Epithelial | Malignant Epithelial | Fibroblast | Muscle | Endothelial |
> | ------------------------- | ---------- | ----- | ------------ | ------------------ | -------------------- | ---------- | ------ | ----------- |
> | UNet                      | 91.94      | 0.87  | 37.03        | 16.37              | 58.67                | 18.73      | 25.41  | 0.52        |
> | UNet + $\mathcal{L}_{fd}$ | 91.77      | 1.82  | 34.59        | 32.93              | 59.53                | 28.18      | 36.48  | 1.69        |
>
> ---
>
> ### **@ Missing baselines SIFA**
>
> We clarify that approaches like **SIFA** were not included because *they address a different problem setting*. Methods such as SIFA assume a **source–target paradigm with known domain labels** and rely on **adversarial feature alignment** to match global distributions.
>
> In contrast, our work focuses on **data-scarce pooling scenarios where datasets are incrementally added**, domain boundaries may be ambiguous, and **no source–target distinction or domain labels are assumed**. Forcing such methods into our setting would require introducing artificial source–target splits and domain labels, fundamentally altering the problem formulation and leading to an unfair comparison.
>
> Our method does not perform adversarial alignment or global feature matching. Instead, it enforces **structured foreground–background disentanglement within each dataset and symmetrically across datasets**, which empirically stabilizes training under data addition without collapsing dataset-specific appearance information.
>
> If the reviewer suggests, we will add a discussion in the camera-ready version positioning our approach relative to adversarial domain-adaptation methods.
>
> ---
>
> ### **@ Large distribution shift (CT vs MRI)**
>
> Adapting between fundamentally different imaging modalities such as **CT and MRI** constitutes a **cross-modality domain adaptation** problem, which is outside the scope of the present work.
>
> Our study focuses on the **Data Addition Dilemma in same-modality, multi-source pooling**, where datasets share the same semantic task and imaging physics but differ in acquisition protocols, scanners, or site-specific appearance statistics. In this regime, foreground–background semantics are consistent and exchangeability of semantic roles is a reasonable assumption.
>
> In contrast, CT and MRI differ substantially in **physical imaging mechanisms, contrast formation, and anatomical appearance**, leading to shifts that are not limited to appearance statistics but also alter the underlying image formation process. Such scenarios typically require modality-translation or cross-modality domain adaptation techniques, which involve different assumptions and methodological considerations.
>
> ---
> *We sincerely appreciate the reviewer’s thoughtful feedback and support!* We hope that the additional experimental results and the justifications strengthens the reviewer’s overall assessment of the submission.

---

> > ### Author Response · Authors · 2026-01-29
> >
> > Dear reviewer,
> > we have added additional experimentations and justifications to the concerns raised and are eager to engage in a discussion to further resolve any other confusion or concern that needs to be addressed in the paper. Looking forward to your response.

---

### Author Response · Authors · 2026-02-01

We thank the reviewers for their initial reviews. We thank the reviewers for highlighting **the method is theoretically grounded** and **the proposed loss is an easy and effective plug-in approach**. We are also encouraged by the reviewers identifying **the problem statement as an important and highly relevant problem in the medical imaging domain**.

We have provided additional experimentations and explanations to clarify the questions of the reviewers. To summarize:

*1. We have provided additional analysis on the distributional difference vs performance for "Data Addition Dilemma" setting where we see the expected trend of higher distributional difference leads to lower performance. This also explains the peaks and dips in Figure 3.
2. We clarify the assumptions made in frontdoor adjustment and the exchangeability assumption.
3. We provide additional ablation to justify the choice of α.
4. We provide additional experimentation demonstrating the flexibility of the proposed feature discrepancy loss by extending it to multiclass setting.*

We appreciate the time reviewers spent on our paper. We were informed of the Feb 1st deadline for the discussion period. We invite any final discussions and clarifications. If the responses and additional experiments are satisfactory, we invite the reviewers to consider revising their scores.

---

### Meta-Review · Area_Chair_gLov · 2026-02-08

**Recommendation:** Accept (Poster)
**Confidence:** 4

**Metareview:**

The paper received one borderline, one weak accept, and one weak reject before the rebuttal. During the rebuttal period, the authors addressed some of the questions raised by reviewers. As such, the final rating was borderline (updated from weak accept), weak accept (no final rating), and borderline (kept the same after the rebuttal). I feel, authors have made substantial changes and provided clarification to warrant acceptance of the paper. Considering this, I am recommending acceptance of the paper.

---

### Decision · Program_Chairs · 2026-02-13

Accept (Poster)